# HYPERPARAMETER TUNING WITH RENYI DIFFERENTIAL PRIVACY

**Nicolas Papernot**[*]
Google Research, Brain Team
`papernot@google.com`

**Thomas Steinke**[*]
Google Research, Brain Team
`hyper@thomas-steinke.net`

## ABSTRACT

For many differentially private algorithms, such as the prominent noisy stochastic gradient descent (DP-SGD), the analysis needed to bound the privacy leakage of a single training run is well understood. However, few studies have reasoned about the privacy leakage resulting from the multiple training runs needed to fine tune the value of the training algorithm's hyperparameters. In this work, we first illustrate how simply setting hyperparameters based on non-private training runs can leak private information. Motivated by this observation, we then provide privacy guarantees for hyperparameter search procedures within the framework of Renyi Differential Privacy. Our results improve and extend the work of Liu and Talwar (STOC 2019). Our analysis supports our previous observation that tuning hyperparameters does indeed leak private information, but we prove that, under certain assumptions, this leakage is modest, as long as each candidate training run needed to select hyperparameters is itself differentially private.

## 1 INTRODUCTION

Machine learning (ML) systems memorize training data and regurgitate excerpts from it when probed (Carlini et al., 2020). If the training data includes sensitive personal information, then this presents an unacceptable privacy risk (Shokri et al., 2017). It may however still be useful to apply machine learning to such data, e.g., in the case of healthcare (Kourou et al., 2015; Wiens et al., 2019). This has led to a significant body of research on the development of privacy-preserving machine learning methods. Differential privacy (DP) (Dwork et al., 2006b;a) provides a robust and quantitative privacy guarantee. It has been widely accepted as the best framework for formally reasoning about the privacy guarantees of a machine learning algorithm.

A popular method for ensuring DP is noisy (stochastic) gradient descent (a.k.a. DP-SGD) (Song et al., 2013; Bassily et al., 2014; Abadi et al., 2016). DP-SGD differs from standard (stochastic) gradient descent in three ways. First, gradients are computed on a per-example basis rather than directly averaged across a minibatch of training examples. Second, each of these individual gradients is clipped to ensure its 2-norm is bounded. Third, Gaussian noise is added to the gradients as they are averaged and applied to update model parameters. These modifications bound the sensitivity of each update so that the added noise ensures differential privacy. The composition (Dwork et al., 2010) and privacy amplification by subsampling (Balle et al., 2018) properties of differential privacy thus imply that the overall training procedure is differentially private. We can compute tight privacy loss bounds for DP-SGD using techniques like the Moments Accountant (Abadi et al., 2016) or the closely related framework of Rényi DP (Mironov, 2017; Mironov et al., 2019).

Machine learning systems have hyperparameters, such as the learning rate, minibatch size, or choice of a regularizer to prevent overfitting. Details of the model architecture can also be treated as hyperparameters of the optimization problem. Furthermore, learning within the constraints of differential privacy may introduce additional hyperparameters, as illustrated in the DP-SGD optimizer by the 2-norm bound value for clipping, the scale of the Gaussian noise, and the choice of stopping time. Typically the training procedure is repeated many times with different hyperparameter settings in order to select the best setting, an operation known as hyperparameter *tuning*. This methodology implies that even if each run of the training procedure is privacy-preserving, we need to take into

---

[*]Alphabetical author order.

account the fact that the training procedure is repeated (possibly many times) when reasoning about the privacy of the overall learning procedure.

*Can the tuning of hyperparameters reveal private information?* This question has received remarkably little attention and, in practice, it is often ignored entirely. We study this question and provide both positive and negative answers.

## 1.1 OUR CONTRIBUTIONS

- **We show that, under certain circumstances, the setting of hyperparamters can leak private information.** Hyperparameters are a narrow channel for private information to leak through, but they can still reveal information about individuals if we are careless. Specifically, if we tune the hyperparameters in an entirely non-private fashion, then individual outliers can noticeably skew the optimal hyperparameter settings. This is sufficient to reveal the presence or absence of these outliers à la membership inference (Shokri et al., 2017); it shows that we must exercise care when setting hyperparameters.

- **We provide tools for ensuring that the selection of hyperparameters is differentially private.** Specifically, if we repeat the training procedure multiple times (with different hyperparameters) and each repetition of the training procedure is differentially private on its own, then outputting the best repetition is differentially private. Of course, a basic version of such a result follows from the composition properties of differential privacy (that is the fact that one can "sum" the privacy loss bounds of multiple differentially private analyses performed on the same data to bound the overall privacy loss from analyzing this data). However, we provide quantitatively sharper bounds.

  Specifically, our privacy loss bounds are either independent of the number of repetitions or grow logarithmically in the number of repetitions, whereas composition would give linear bounds. Rather than repeating the training procedure a fixed number of times, our results require repeating the training procedure a random number of times. The privacy guarantees depend on the distribution of the number of runs; we consider several distributions and provide generic results. We discover a tradeoff between the privacy parameters and how heavy-tailed the distribution of the number of repetitions is.

## 1.2 BACKGROUND AND RELATED WORK

Differential privacy (DP) is a framework to reason about the privacy guarantees of randomized algorithms which analyze data (Dwork et al., 2006b;a). An algorithm is said to be DP if its outputs on any pair of datasets that only differ on one individual's data are indistinguishable. A bound on this indistinguishability serves as the quantification for privacy. Formally, a randomized algorithm $M : \mathcal{X}^n \to \mathcal{Y}$ is $(\varepsilon, \delta)$-DP if for any inputs $x, x' \in \mathcal{X}^n$ differing only on the addition, removal, or replacement of one individual's records and for any subset of outputs $S \subseteq \mathcal{Y}$, we have

$$\mathbb{P}\left[M(x) \in S\right] \leq e^{\varepsilon} \mathbb{P}\left[M(x') \in S\right] + \delta. \tag{1}$$

Here, the parameter $\varepsilon$ is known as the privacy loss bound – the smaller $\varepsilon$ is, the stronger the privacy guarantee provided is, because it is hard for an adversary to distinguish the outputs of the algorithm on two adjacent inputs. The parameter $\delta$ is essentially the probability that the guarantee fails to hold. One of the key properties of DP is that it composes: running multiple independent DP algorithms is also DP and composition theorems allow us to bound the privacy parameters of such a sequence of mechanisms in terms of the individual mechanisms' privacy parameters (Dwork & Roth, 2014).

There is a vast literature on differential privacy in machine learning. A popular tool is the DP-SGD optimizer (Abadi et al., 2016). Because the noise added is Gaussian and DP-SGD applies the same (differentially private) training step sequentially, it is easier to reason about its privacy guarantees in the framework of Rényi differential privacy (Mironov, 2017). Rényi differential privacy (RDP) generalizes pure differential privacy (with $\delta = 0$) as follows. An algorithm $M$ is said to be $(\lambda, \varepsilon)$-RDP with $\lambda \geq 1$ and $\varepsilon \geq 0$, if for any adjacent inputs $x, x'$

$$\mathrm{D}_\lambda \left(M(x) \| M(x')\right) := \frac{1}{\lambda - 1} \log \mathop{\mathbb{E}}_{Y \leftarrow M(x)} \left[ \left( \frac{\mathbb{P}\left[M(x) = Y\right]}{\mathbb{P}\left[M(x') = Y\right]} \right)^{\lambda - 1} \right] \leq \varepsilon, \tag{2}$$

where $\mathrm{D}_\lambda \left(P \| Q\right)$ is the Rényi divergence of order $\lambda$ between distributions $P$ and $Q$. In the framework of RDP, one obtains sharp and simple composition: If each individual mechanism $M_i$ is $(\lambda, \varepsilon_i)$-RDP,

then the composition of running all of the mechanisms on the data satisfies $(\lambda, \sum_i \varepsilon_i)$-RDP. For instance, the privacy analysis of DP-SGD first analyzes the individual training steps then applies composition. Note that it is common to keep track of multiple orders $\lambda$ in the analysis. Thus $\varepsilon$ should be thought of as a function $\varepsilon(\lambda)$, rather than a single number. In many cases, such as Gaussian noise addition, this is a linear function – i.e., $\varepsilon(\lambda) = \rho \cdot \lambda$ for some $\rho \in \mathbb{R}$ – and such a linear bound yields the definition of zero-Concentrated DP with parameter $\rho$ ($\rho$-zCDP) (Bun & Steinke, 2016).

One could naïvely extend this composition-based approach to analyze the privacy of a training algorithm which involves hyperparameter tuning. Indeed, if each training run performed to evaluate one candidate set of hyperparameter values is DP, the overall procedure is also DP by composition over all the hyperparameter values tried. However, this would lead to very loose guarantees of privacy. Chaudhuri & Vinterbo (2013) were the first to obtain improved DP bounds for hyperparameter tuning, but their results require a stability property of the learning algorithm. The only prior work that has attempted to obtain tighter guarantees for DP hyperparameter tuning in a black-box fashion is the work of Liu & Talwar (2019). Their work is the starting point for ours.

Liu & Talwar (2019) show that, if we start with a $(\varepsilon, 0)$-DP algorithm, repeatedly run it a random number of times following a geometric distribution, and finally return the best output produced by these runs, then this system satisfies $(3\varepsilon, 0)$-differential privacy.[1] Liu & Talwar (2019) also consider algorithms satisfying $(\varepsilon, \delta)$-DP for $\delta > 0$. However, their analysis is restricted to the $(\varepsilon, \delta)$ formulation of DP and they do not give any results for Rényi DP. This makes it difficult to apply these results to modern DP machine learning systems, such as models trained with DP-SGD.

Our results directly improve on the results of Liu & Talwar (2019). We show that replacing the geometric distribution on the number of repetitions in their result with the logarithmic distribution yields $(2\varepsilon, 0)$-differential privacy as the final result. We also consider other distributions on the number of repetitions, which give a spectrum of results. We simultaneously extend these results to the Rényi DP framework, which yields sharper privacy analyses.

Independently Mohapatra et al. (2021) study *adaptive* hyperparameter tuning under DP with composition. In contrast, our results are for non-adaptive hyperparameter tuning, i.e., "random search."

A closely related line of work is on the problem of private selection. Well-known algorithms for private selection include the exponential mechanism (McSherry & Talwar, 2007) and the sparse vector technique (Dwork et al., 2009; Zhu & Wang, 2020). However, this line of work assumes that there is a low-sensitivity function determining the quality of each of the options. This is usually not the case for hyperparameters. Our results simply treat the ML algorithm as a black box; we only assume that its output is private and make no assumptions about how that output was generated. Our results also permit returning the entire trained model along with the selected hyperparameters.

## 2  MOTIVATION

A hyperparameter typically takes categorical values (e.g., the choice of activation function in a neural network layer), or is a single number (e.g., a real number for the learning rate or an integer for the number of epochs). Thus, it is intuitive that a hyperparameter provides little capacity as a channel to leak private information from the training data. Nevertheless, leakage can happen, in particular when training is done without preserving privacy. We illustrate how with the constructed example of hyperparameter tuning for a support vector machine learning (SVM) learned from a synthetic data distribution. We consider a SVM with a soft margin; we use stochastic gradient descent to minimize the corresponding objective involving the hinge loss and a weight penalty:

$$l_{w,b}(x, y) = \|w\|_2^2 + \alpha \max\{0, 1 - y(w \cdot x + b)\}$$

where $y \in \{-1, 1\}$ indicates the label of training example $x \in \mathbb{R}^2$. Because our purpose here is to illustrate how leakage of private information can arise from hyperparameter tuning, we work with a synthetic data distribution for simplicity of exposition: we draw 40 inputs from isotropic 2D Gaussians of standard deviation 1.0 to form the training set $\mathcal{D}$. The negative class is sampled from a Gaussian centered at $\mu_{-1} = (7.86, -3.36)$ and the positive at $\mu_1 = (6.42, -9.17)$.

Our learning procedure has a single hyperparameter $\alpha$ controlling how much importance is given to the hinge loss, i.e., how much the SVM is penalized for using slack variables to misclassify some of

---

[1]Liu & Talwar (2019) prove several other results with slightly different formulations of the problem, but this result is representative and is most relevant to our discussion here.

the training data. We first tune the value of $\alpha$ with the training set $\mathcal{D}$ and report training accuracy as a function of $\alpha$ in Figure 1. Next, we repeat this experiment on a dataset $\mathcal{D}'$ to which we added 8 outliers $x_o = (7.9, -8.0)$ to the negative class. The resulting hyperparameter tuning curve is added to Figure 1. By comparing both curves, it is clear that the choice of hyperparameter $\alpha$ which maximizes accuracy differs in the two settings: the best performance is achieved around $\alpha = 8$ with outliers whereas increasing $\alpha$ is detrimental to performance without outliers.

This difference can be exploited to perform a variant of a membership inference attack (Shokri et al., 2017): Here, one could infer from the value of the hyperparameter $\alpha$ whether or not the outlier points $x_o$ were part of the training set or not. While the example is constructed, this shows how we must be careful when tuning hyperparameters: in corner cases such as the one presented here, it is possible for some information contained in the training data to leak in hyperparameter choices.

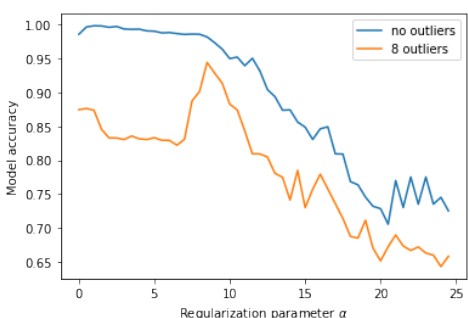

Figure 1: Accuracy of the model as a function of the regularization weight $\alpha$, with and without outliers. Note how the model performance exhibits a turning point with outliers whereas increasing the value of $\alpha$ is detrimental without outliers.

In particular, this implies that the common practice of tuning hyperparameters without differential privacy and then using the hyperparameter values selected to repeat training one last time with differential privacy is not ideal. In Section 3, we will in particular show how training with differential privacy when performing the different runs necessary to tune the hyperparameter can bound such leakage effectively if one carefully chooses the number of runs hyperparameters are tuned for.

## 3 OUR POSITIVE RESULTS

### 3.1 PROBLEM FORMULATION

We begin by appropriately formalizing the problem of differentially private hyperparameter tuning, following the framework of Liu & Talwar (2019). Suppose we have $m$ randomized base algorithms $M_1, M_2, \cdots, M_m : \mathcal{X}^n \to \mathcal{Y}$. These correspond to $m$ possible settings of the hyperparameters. Ideally we would simply run each of these algorithms once and return the best outcome. For simplicity, we consider a finite set of hyperparameter possibilities; if the hyperparameters of interest are continuous, then we must pick a finite subset to search over (which is in practice sufficient).

Here we make two simplifying assumptions: First, we assume that there is a total order on the range $\mathcal{Y}$, which ensures that "best" is well-defined. In particular, we are implicitly assuming that the algorithm computes a quality score (e.g., accuracy on a test set) for the trained model it produces; this may require allocating some privacy budget to this evaluation.[2] Second, we are assuming that the output includes both the trained model and the corresponding hyperparameter values (i.e., the output of $M_j$ includes the index $j$).[3] These assumptions can be made without loss of generality.

### 3.2 STRAWMAN APPROACH: REPEAT THE BASE ALGORITHM A FIXED NUMBER OF TIMES

The obvious approach to this problem would be to run each algorithm once and to return the best of the $m$ outcomes. From composition, we know that the privacy cost grows at most linearly with $m$. It turns out that this is in fact tight. There exists a $(\varepsilon, 0)$-DP algorithm such that if we repeatedly run it $m$ times and return the best output, the resultant procedure is *not* $(m\varepsilon - \tau, 0)$-DP for any $\tau > 0$. This was observed by Liu & Talwar (2019, Appendix B) and we provide an analysis in Appendix D.1. This negative result also extends to Rényi DP. To avoid this problem, we will run the base algorithms a random number of times. The added uncertainty significantly enhances privacy. However, we must carefully choose this random distribution and analyze it.

---

[2]If the trained model's quality is evaluated on the training set, then we must increase the privacy loss budget to account for this composition. However, if the model is evaluated on a held out set, then the privacy budget need not be split; these data points are "fresh" from a privacy perspective.

[3]Note that in the more well-studied problem of selection, usually we would only output the hyperparameter values (i.e., just the index $j$) and not the corresponding trained model nor its quality score.

### 3.3 Our algorithm for hyperparameter tuning

To obtain good privacy bounds, we must run the base algorithms a random number of times. We remark that random search rather than grid search is often performed in practice (Bergstra & Bengio, 2012), so this is not a significant change in methodology. Specifically, we pick a total number of runs $K$ from some distribution. Then, for each run $k = 1, 2, \cdots, K$, we pick an index $j_k \in [m]$ uniformly at random and run $M_{j_k}$. Then, at the end, we return the best of the $K$ outcomes.

The privacy guarantee of this overall system then depends on the privacy guarantees of each of the mechanisms $M_j$ as well as the distribution of the number of runs $K$. Specifically, we assume that there exists a uniform (Rényi) DP bound for all of the mechanisms $M_j$. Note that DP is "convex" where "convexity" here means that if $M_1, M_2, \cdots, M_m$ are each individually DP, then running $M_{j_k}$ for a random $j_k \in [m]$ is also DP with the same parameters.

To simplify notation, we also assume that there is a single mechanism $Q : \mathcal{X}^n \to \mathcal{Y}$ that picks a random index $j \in [m]$ and then runs $M_j$. In essence, our goal is to "boost" the success probability of $Q$ by repeating it many times. The distribution of the number of runs $K$ must be chosen to both ensure good privacy guarantees and to ensure that the system is likely to pick a good setting of hyperparameters. Also, the overall runtime of the system depends on $K$ and we want the runtime to be efficient and predictable. Our results consider several distributions on the number of repetitions $K$ and the ensuing tradeoff between these three considerations.

### 3.4 Main Results

There are many possible distributions for the number of repetitions $K$. In this section, we first consider two – the truncated negative binomial and Poisson distributions – and state our main privacy results for these distributions. Later, in Section 3.5, we state a more general technical lemma which applies to any distribution on the number of repetitions $K$.

**Definition 1** (Truncated Negative Binomial Distribution). *Let $\gamma \in (0, 1)$ and $\eta \in (-1, \infty)$. Define a distribution $\mathcal{D}_{\eta,\gamma}$ on $\mathbb{N} = \{1, 2, \cdots\}$ as follows. If $\eta \neq 0$ and $K$ is drawn from $\mathcal{D}_{\eta,\gamma}$, then*

$$\forall k \in \mathbb{N} \quad \mathbb{P}\left[K = k\right] = \frac{(1-\gamma)^k}{\gamma^{-\eta} - 1} \cdot \prod_{\ell=0}^{k-1} \left(\frac{\ell + \eta}{\ell + 1}\right) \tag{3}$$

*and $\mathbb{E}\left[K\right] = \frac{\eta \cdot (1-\gamma)}{\gamma \cdot (1-\gamma^\eta)}$. If $K$ is drawn from $\mathcal{D}_{0,\gamma}$, then*

$$\mathbb{P}\left[K = k\right] = \frac{(1-\gamma)^k}{k \cdot \log(1/\gamma)} \tag{4}$$

*and $\mathbb{E}\left[K\right] = \frac{1/\gamma - 1}{\log(1/\gamma)}$.*

This is called the "negative binomial distribution," since $\prod_{\ell=0}^{k-1}\left(\frac{\ell+\eta}{\ell+1}\right) = \binom{k+\eta-1}{k}$ if we extend the definition of binomial coefficients to non-integer $\eta$. The distribution is called "truncated" because $\mathbb{P}\left[K = 0\right] = 0$, whereas the standard negative binomial distribution includes 0 in its support. The $\eta = 0$ case $\mathcal{D}_{0,\gamma}$ is known as the "logarithmic distribution". The $\eta = 1$ case $\mathcal{D}_{1,\gamma}$ is simply the geometric distribution. Next, we state our main privacy result for this distribution.

**Theorem 2** (Main Privacy Result – Truncated Negative Binomial). *Let $Q : \mathcal{X}^n \to \mathcal{Y}$ be a randomized algorithm satisfying $(\lambda, \varepsilon)$-RDP and $(\hat{\lambda}, \hat{\varepsilon})$-RDP for some $\varepsilon, \hat{\varepsilon} \geq 0$, $\lambda \in (1, \infty)$, and $\hat{\lambda} \in [1, \infty)$.[4] Assume $\mathcal{Y}$ is totally ordered.*

*Let $\eta \in (-1, \infty)$ and $\gamma \in (0, 1)$. Define an algorithm $A : \mathcal{X}^n \to \mathcal{Y}$ as follows. Draw $K$ from the truncated negative binomial distribution $\mathcal{D}_{\eta,\gamma}$ (Definition 1). Run $Q(x)$ repeatedly $K$ times. Then $A(x)$ returns the best value from the $K$ runs.*

*Then $A$ satisfies $(\lambda, \varepsilon')$-RDP where*

$$\varepsilon' = \varepsilon + (1 + \eta) \cdot \left(1 - \frac{1}{\hat{\lambda}}\right)\hat{\varepsilon} + \frac{(1+\eta) \cdot \log(1/\gamma)}{\hat{\lambda}} + \frac{\log \mathbb{E}\left[K\right]}{\lambda - 1}. \tag{5}$$

---

[4]If $\hat{\lambda} = 1$, then $\hat{\varepsilon}$ corresponds to the KL divergence; see Definition 9. However, $\hat{\varepsilon}$ is multiplied by $1 - 1/\hat{\lambda} = 0$ in this case, so is irrelevant.

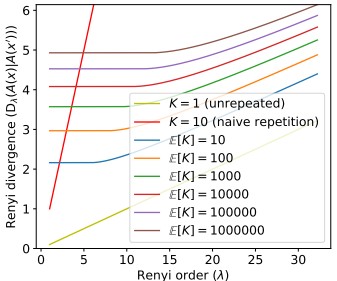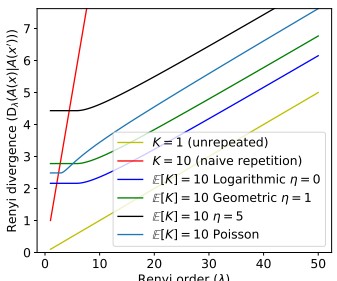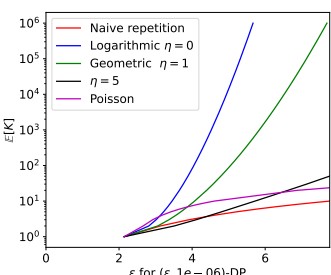

Figure 2: Rényi DP guarantees from Corollary 4 for various expected numbers of repetitions of the logarithmic distribution (i.e., truncated negative binomial with $\eta = 0$), compared with base algorithm (0.1-zCDP) and naïve composition.

Figure 3: Rényi DP guarantees for repetition using different distributions or naïve composition with mean 10, with 0.1-zCDP base algorithm.

Figure 4: Privacy versus expected number of repetitions using different distributions or naïve composition. Rényi DP guarantees are converted to approximate DP – i.e., we plot $\varepsilon$ such that we attain $(\varepsilon, 10^{-6})$-DP. The base algorithm is 0.1-zCDP.

Theorem 2 shows a tradeoff between privacy and utility for the distribution $\mathcal{D}_{\eta,\gamma}$ of the number of repetitions. Privacy improves as $\eta$ decreases and $\gamma$ increases. However, this corresponds to fewer repetitions and thus a lower chance of success. We will study this aspect in Section 3.6.

Theorem 2 assumes two RDP bounds, which makes it slightly hard to interpret. Thus we consider two illustrative special cases: We start with pure DP (a.k.a. pointwise DP) – i.e., $(\varepsilon, \delta)$-DP with $\delta = 0$, which is equivalent to $(\infty, \varepsilon)$-RDP. This corresponds to Theorem 2 with $\lambda \to \infty$ and $\hat{\lambda} \to \infty$.

**Corollary 3** (Theorem 2 for pure DP). *Let $Q : \mathcal{X}^n \to \mathcal{Y}$ be a randomized algorithm satisfying $(\varepsilon, 0)$-DP. Let $\eta \in (-1, \infty)$ and $\gamma \in (0, 1)$. Define $A : \mathcal{X}^n \to \mathcal{Y}$ as in Theorem 2. Then $A$ satisfies $\big((2 + \eta)\varepsilon, 0\big)$-DP.*

Our result is a generalization of the result of Liu & Talwar (2019) – they show that, if $K$ follows a geometric distribution and $Q$ satisfies $(\varepsilon, 0)$-DP, then $A$ satisfies $(3\varepsilon, 0)$-DP. Setting $\eta = 1$ in Corollary 3 recovers their result. If we set $\eta < 1$, then we obtain an improved privacy bound.

Another example is if $Q$ satisfies concentrated DP (Dwork & Rothblum, 2016; Bun & Steinke, 2016). This is the type of guarantee that is obtained by adding Gaussian noise to a bounded sensitivity function. In particular, this is the type of guarantee we would obtain from noisy gradient descent.[5]

**Corollary 4** (Theorem 2 for Concentrated DP). *Let $Q : \mathcal{X}^n \to \mathcal{Y}$ be a randomized algorithm satisfying $\rho$-zCDP – i.e., $(\lambda, \rho \cdot \lambda)$-Rényi DP for all $\lambda > 1$. Let $\eta \in (-1, \infty)$ and $\gamma \in (0, 1)$. Define $A : \mathcal{X}^n \to \mathcal{Y}$ and $K \leftarrow \mathcal{D}_{\eta,\gamma}$ as in Theorem 2. Assume $\rho \leq \log(1/\gamma)$. Then $A$ satisfies $(\lambda, \varepsilon')$-Rényi DP for all $\lambda > 1$ with*

$$
\varepsilon' = \begin{cases} 2\sqrt{\rho \cdot \log\left(\mathbb{E}\left[K\right]\right)} + 2(1+\eta)\sqrt{\rho \log(1/\gamma)} - \eta\rho & \text{if } \lambda \leq 1 + \sqrt{\frac{1}{\rho}\log\left(\mathbb{E}\left[K\right]\right)} \\ \rho \cdot (\lambda - 1) + \frac{1}{\lambda-1}\log\left(\mathbb{E}\left[K\right]\right) + 2(1+\eta)\sqrt{\rho \log(1/\gamma)} - \eta\rho & \text{if } \lambda > 1 + \sqrt{\frac{1}{\rho}\log\left(\mathbb{E}\left[K\right]\right)} \end{cases} .
$$

Figure 2 shows what the guarantee of Corollary 4 looks like. Here we start with 0.1-zCDP and perform repetition following the logarithmic distribution ($\eta = 0$) with varying scales (given by $\gamma$) and plot the Rényi DP guarantee attained by outputting the best of the repeated runs. The improvement over naive composition, which instead grows linearly, is clear. We also study other distributions on the number of repetitions, obtained by varying $\eta$, and Figure 3 gives a comparison. Figure 4 shows what these bounds look like if we convert to approximate $(\varepsilon, \delta)$-DP with $\delta = 10^{-6}$.

**Remark 5.** *Corollary 4 uses the monotonicity property of Rényi divergences: If $\lambda_1 \leq \lambda_2$, then $\mathrm{D}_{\lambda_1}\left(P\|Q\right) \leq \mathrm{D}_{\lambda_2}\left(P\|Q\right)$ (Van Erven & Harremos, 2014, Theorem 3). Thus $(\lambda_2, \varepsilon)$-RDP implies $(\lambda_1, \varepsilon)$-RDP for any $\lambda_1 \leq \lambda_2$. In particular, the bound of Theorem 2 yields $\varepsilon' \to \infty$ as $\lambda \to 1$, so we use monotonicity to bound $\varepsilon'$ for small $\lambda$.*

---

[5]Note that the privacy of noisy *stochastic* gradient descent (DP-SGD) is not well characterized by concentrated DP (Bun et al., 2018), but, for our purposes, this is a close enough approximation.

**Poisson Distribution.** We next consider the Poisson distribution, which offers a different privacy-utility tradeoff than the truncated negative binomial distribution. The Poisson distribution with mean $\mu \geq 0$ is given by $\mathbb{P}[K = k] = e^{-\mu}\frac{\mu^k}{k!}$ for all $k \geq 0$. Note that $\mathbb{P}[K = 0] = e^{-\mu} > 0$ here, whereas the truncated negative binomial distribution does not include 0 in its support. We could condition on $K \geq 1$ here too, but we prefer to stick with the standard definition. We remark that (modulo the issue around $\mathbb{P}[K = 0]$) the Poisson distribution is closely related to the truncated negative binomial distribution. If we take the limit as $\eta \to \infty$ while the mean remains fixed, then the negative binomial distribution becomes a Poisson distribution. Conversely, the negative binomial distribution can be represented as a convex combination of Poisson distributions or as a compound of Poisson and logarithmic; see Appendix A.2 for more details.

**Theorem 6** (Main Privacy Result – Poisson Distribution). *Let $Q : \mathcal{X}^n \to \mathcal{Y}$ be a randomized algorithm satisfying $(\lambda, \varepsilon)$-RDP and $(\hat{\varepsilon}, \hat{\delta})$-DP for some $\lambda \in (1, \infty)$ and $\varepsilon, \hat{\varepsilon}, \hat{\delta} \geq 0$. Assume $\mathcal{Y}$ is totally ordered. Let $\mu > 0$.*

*Define an algorithm $A : \mathcal{X}^n \to \mathcal{Y}$ as follows. Draw $K$ from a Poisson distribution with mean $\mu$ – i.e., $\mathbb{P}[K = k] = e^{-\mu} \cdot \frac{\mu^k}{k!}$ for all $k \geq 0$. Run $Q(x)$ repeatedly $K$ times. Then $A(x)$ returns the best value from the $K$ runs. If $K = 0$, $A(x)$ returns some arbitrary output independent from the input $x$. If $e^{\hat{\varepsilon}} \leq 1 + \frac{1}{\lambda - 1}$, then $A$ satisfies $(\lambda, \varepsilon')$-RDP where*

$$\varepsilon' = \varepsilon + \mu \cdot \hat{\delta} + \frac{\log \mu}{\lambda - 1}.$$

The assumptions of Theorem 6 are different from Theorem 2: We assume a Rényi DP guarantee and an approximate DP guarantee on $Q$, rather than two Rényi DP guarantees. We remark that a Rényi DP guarantee can be converted into an approximate DP guarantee – $(\lambda, \varepsilon)$-RDP implies $(\hat{\varepsilon}, \hat{\delta})$-DP for all $\hat{\varepsilon} \geq \varepsilon$ and $\hat{\delta} = e^{(\lambda-1)(\hat{\varepsilon}-\varepsilon)} \cdot \frac{1}{\lambda} \cdot \left(1 - \frac{1}{\lambda}\right)^{\lambda-1}$ (Mironov, 2017; Canonne et al., 2020). Thus this statement can be directly compared to our other result. We show such a comparison in Figure 3 and Figure 4. The proofs of Theorems 2 and 6 are included in Appendix B.2.

### 3.5 GENERIC RÉNYI DP BOUND FOR ANY DISTRIBUTION ON THE NUMBER OF REPETITIONS

We now present our main technical lemma, which applies to any distribution on the number of repetitions $K$. Theorems 2 and 6 are derived from this result. It gives a Rényi DP bound for the repeated algorithm in terms of the Rényi DP of the base algorithm and the probability generating function of the number of repetitions applied to probabilities derived from the base algorithm.

**Lemma 7** (Generic Bound). *Fix $\lambda > 1$. Let $K$ be a random variable supported on $\mathbb{N} \cup \{0\}$. Let $f : [0, 1] \to \mathbb{R}$ be the probability generating function of $K$ – i.e., $f(x) := \sum_{k=0}^{\infty} \mathbb{P}[K = k] \cdot x^k$.*

*Let $Q$ and $Q'$ be distributions on $\mathcal{Y}$. Assume $\mathcal{Y}$ is totally ordered. Define a distribution $A$ on $\mathcal{Y}$ as follows. First sample $K$. Then sample from $Q$ independently $K$ times and output the best of these samples.[6] This output is a sample from $A$. We define $A'$ analogously with $Q'$ in place of $Q$. Then*

$$\mathrm{D}_\lambda (A \| A') \leq \mathrm{D}_\lambda (Q \| Q') + \frac{1}{\lambda - 1} \log \left( f'(q)^\lambda \cdot f'(q')^{1-\lambda} \right), \tag{6}$$

*where applying the same postprocessing to $Q$ and $Q'$ gives probabilities $q$ and $q'$ respectively – i.e., there exists an arbitrary function $g : \mathcal{Y} \to [0, 1]$ such that $q = \underset{X \leftarrow Q}{\mathbb{E}}[g(X)]$ and $q' = \underset{X' \leftarrow Q'}{\mathbb{E}}[g(X')]$.*

The proof of this generic bound is found in Appendix B.1. To interpret the theorem, we should imagine adjacent inputs $x, x' \in \mathcal{X}^n$, and then the distributions correspond to the algorithms run on these inputs: $A = A(x)$, $A' = A(x')$, $Q = Q(x)$, and $Q' = Q(x')$. The bounds on Rényi divergence thus correspond to Rényi DP bounds. The derivative of the probability generating function – $f'(x) = \mathbb{E}\left[K \cdot x^{K-1}\right]$ – is somewhat mysterious. A first-order intuition is that, if $q = q'$, then $f'(q)^\lambda \cdot f'(q')^{1-\lambda} = f'(q) \leq f'(1) = \mathbb{E}[K]$ and thus the last term in the bound (6) is simply $\frac{\log \mathbb{E}[K]}{\lambda - 1}$. A second-order intuition is that $q \approx q'$ by DP and postprocessing and, if $f'$ is smooth, then $f'(q) \approx f'(q')$ and the first-order intuition holds up to these approximations. Vaguely, $f'$ being smooth corresponds to the distribution of $K$ being spread out (i.e., not a point mass) and not too

---

[6]If $K = 0$, the output can be arbitrary, as long as it is the same for both $A$ and $A'$.

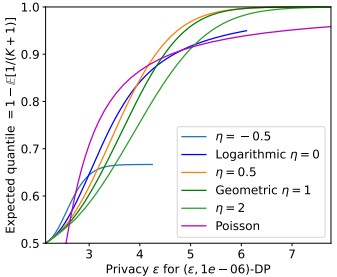 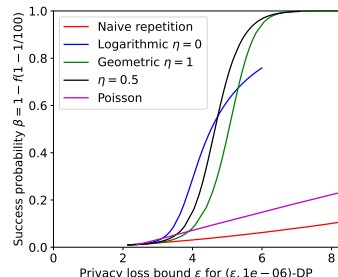 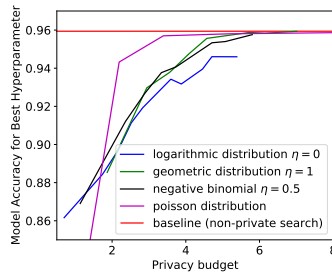

Figure 5: Expected quantile of the repeated algorithm $A$ as a function of the final privacy guarantee $(\varepsilon, 10^{-6})$-DP for various distributions $K$, where each invocation of the base algorithm $Q$ is $0.1$-zCDP.

Figure 6: Final success probability ($\beta$) of the repeated algorithm $A$ as a function of the final privacy guarantee $(\varepsilon, 10^{-6})$-DP for various distributions, where each invocation of the base algorithm $Q$ has a $1/100$ probability of success and is $0.1$-zCDP.

Figure 7: Accuracy of the CNN model obtained at the end of the hyperparameter search, for the different distributions on the number of repetitions $K$ we considered. We report the mean over 500 trials of the experiment.

heavy-tailed (i.e. $K$ is small most of the time). The exact quantification of this smoothness depends on the form of the DP guarantee $q \approx q'$.

In our work, we primarily compare three distributions on the number of repetitions: a point mass (corresponding to naïve repetition), the truncated negative binomial distribution, and the Poisson distribution. A point mass would have a polynomial as the probability generating function – i.e., if $\mathbb{P}[K = k] = 1$, then $f(x) = \mathbb{E}[x^K] = x^k$. The probability generating function of the truncated negative binomial distribution (Definition 1) is

$$f(x) = \mathop{\mathbb{E}}_{K \leftarrow \mathcal{D}_{\eta,\gamma}}[x^K] = \begin{cases} \frac{(1-(1-\gamma)x)^{-\eta}-1}{\gamma^{-\eta}-1} & \text{if } \eta \neq 0 \\ \frac{\log(1-(1-\gamma)x)}{\log(\gamma)} & \text{if } \eta = 0 \end{cases} . \tag{7}$$

The probability generating function of the Poisson distribution with mean $\mu$ is given by $f(x) = e^{\mu \cdot (x-1)}$. We discuss probability generating functions further in Appendix A.2.

### 3.6 UTILITY AND RUNTIME OF OUR HYPERPARAMETER TUNING ALGORITHM

Our analytical results thus far, Theorems 2 and 6 and Lemma 7, provide privacy guarantees for our hyperparameter tuning algorithm $A$ when it is used with various distributions on the number of repetitions $K$. We now turn to the utility that this algorithm provides. The utility of a hyperparameter search is determined by how many times the base algorithm (denoted $Q$ in the theorem statements) is run when we invoke the overall algorithm ($A$). The more often $Q$ is run, the more likely we are to observe a good output and $A$ is more likely to return the corresponding hyperparameter values. Note that the number of repetitions $K$ also determines the algorithm's runtime, so these are closely linked.

*How does this distribution on the number of repetitions $K$ map to utility?* As a first-order approximation, the utility and runtime are proportional to $\mathbb{E}[K]$. Hence several of our figures compare the different distributions on $K$ based on a fixed expectation and Figure 4 plots $\mathbb{E}[K]$ on the vertical axis. However, this first-order approximation ignores the fact that some of the distributions we consider are more concentrated than others; even if the expectation is large, there might still be a significant probability that $K$ is small. Indeed, for $\eta \leq 1$, the mode of the truncated negative binomial distribution is $K = 1$. We found this to be an obstacle to using the (truncated) negative binomial distribution in practice in our experiments, and discuss this further in Appendix A.2.1.

We can formulate utility guarantees more precisely by looking at the expected quantile of the output to measure our algorithm's utility. If we run the base algorithm $Q$ once, then the quantile of the output is (by definition) uniform on $[0, 1]$ and has mean $0.5$. If we repeat the base algorithm a fixed number of times $k$, then the quantile of the best output follows a $\mathsf{Beta}(k, 1)$ distribution, as it is the maximum of $k$ independent uniform random variables. The expectation in this case is $\frac{k}{k+1} = 1 - \frac{1}{k+1}$. If we

repeat a random number of times $K$, the expected quantile of the best result is given by

$$\mathbb{E}\left[\frac{K}{K+1}\right] = \mathbb{E}\left[1 - \frac{1}{K+1}\right] = \int_0^1 x \cdot f'(x)\mathrm{d}x = 1 - \int_0^1 f(x)\mathrm{d}x, \qquad (8)$$

where $f(x) = \mathbb{E}\left[x^K\right]$ is the probability generating function of $K$; see Appendix A.2.1 for further details. Figure 5 plots this quantity against privacy. We see that the Poisson distribution performs very well in an intermediate range, while the negative binomial distribution with $\eta = 0.5$ does well if we want a strong utility guarantee. This means that Poisson is best used when little privacy budget is available for the hyperparameter search. Instead, the negative binomial distribution with $\eta = 0.5$ allows us to improve the utility of the solution returned by the hyperparameter search, but this only holds when spending a larger privacy budget (in our example, the budget has to be at least $\varepsilon = 4$ otherwise Poisson is more advantageous). The negative binomial with $\eta = -0.5$ does very poorly.

From a runtime perspective, the distribution of $K$ should have light tails. All of the distributions we have considered have subexponential tails. However, larger $\eta$ corresponds to better concentration in the negative binomial distribution with the Poisson distribution having the best concentration.

**Experimental Evaluation.** To confirm these findings, we apply our algorithm to a real hyperparameter search task. Specifically, we fine-tune the learning rate of a convolutional neural network trained on MNIST. We implement DP-SGD in JAX for an all-convolutional architecture with a stack of 32, 32, 64, 64, 64 feature maps generated by 3x3 kernels. We vary the learning rate between 0.025 and 1 on a logarithmic scale but fix all other hyperparameters: 60 epochs, minibatch size of 256, $\ell_2$ clipping norm of 1, and noise multiplier of 1.1. In Figure 7, we plot the maximal accuracy achieved during the hyperparameter search for the different distributions considered previously as a function of the total privacy budget expended by the search. The experiment is repeated 500 times and the mean result reported. This experiment shows that the Poisson distribution achieves the best privacy-utility tradeoff for this relatively simple hyperparameter search. This agrees with the theoretical analysis we just presented above that shows that the Poisson distribution performs well in the intermediate range of utility, as this is a simple hyperparameter search.

## 4 CONCLUSION

Our positive results build on the work of Liu & Talwar (2019) and show that repeatedly running the base algorithm and only returning the best output can incur much lower privacy cost than naïve composition would suggest. This however requires that we randomize the number of repetitions, rather than repeating a fixed number of times. We analyze a variety of distributions for the number of repetitions, each of which gives a different privacy/utility tradeoff.

While our results focused on the privacy implications of tuning hyperparameters with, and without, differential privacy, our findings echo prior observations that tuning details of the model architecture without privacy to then repeat training with DP affords suboptimal utility-privacy tradeoffs (Papernot et al., 2020); in this work, the authors demonstrated that the optimal choice of activation function in a neural network can be different when learning with DP, and that tuning it with DP immediately can improve the model's utility at no changes to the privacy guarantee. We envision that future work will be able to build on our algorithm for private tuning of hyperparameters to facilitate privacy-aware searches for model architectures and training algorithm configurations to effectively learn with them.

**Limitations.** We show that hyperparameter tuning is not free from privacy cost. Our theoretical and experimental results show that, in the setting of interest, the privacy parameter may double or even triple after accounting for hyperparameter tuning, which could be prohibitive. In this case, one compromise would be to state both privacy guarantees – that of the base algorithm that does not account for hyperparameter tuning, and that of the overall system that does account for this. The reader may wonder whether our positive results can be improved. In Appendix D, we explore give some intuition for why they cannot (easily) be improved. We also note that our results are only immediately applicable to the hyperparameter tuning algorithm from Section 3.3. Other algorithms, in particular those that adaptively choose hyperparameter candidates will require further analysis.

Finally, among the distributions on the number of repetitions $K$ that we have analyzed, the distribution that provides the best privacy-utility tradeoffs will depend on the setting. While it is good to have choices, this does leave some work to be done by those using our results. Fortunately, the differences between the distributions seem to be relatively small, so this choice is unlikely to be critical.

REPRODUCIBILITY & ETHICS STATEMENTS

**Reproducibility.**   We give precise theorem statements for our main results and we have provided complete proofs in the Appendix, as well as all the necessary calculations and formulas for plotting our figures. We have also fully specified the setup required to reproduce our experimental results, including hyperparameters. Our algorithm is simple, fully specified and can be easily implemented.

**Ethics.**   Our work touches on privacy, which is an ethically sensitive topic. If differentially private algorithms – such as ours – are applied to real-world sensitive data, then potential harms to the people whose data is being used must be carefully considered. However, our work is not directly using real-world sensitive data. Our main results are theoretical and our experiments use either synthetic data or MNIST, which is a standard non-private dataset.

ACKNOWLEGMENTS

The authors would like to thank the reviewers for their detailed feedback and interactive discussion during the review period. We also thank our colleagues Abhradeep Guha Thakurta, Andreas Terzis, Peter Kairouz, and Shuang Song for insightful discussions about differentially private hyperparameter tuning that led to the present project, as well as their comments on early drafts of this document.

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

## A  Further Background

### A.1  Differential Privacy & Rényi DP

For completeness, we provide some basic background on differential privacy and, in particular, Rényi differential privacy. We start with the standard definition of differential privacy:

**Definition 8** (Differential Privacy). *A randomized algorithm $M : \mathcal{X}^n \to \mathcal{Y}$ is $(\varepsilon, \delta)$-differentially private if, for all neighbouring pairs of inputs $x, x' \in \mathcal{X}^n$ and all measurable $S \subset \mathcal{Y}$,*

$$\mathbb{P}\left[M(x) \in S\right] \le e^{\varepsilon} \cdot \mathbb{P}\left[M(x') \in S\right] + \delta.$$

*When $\delta = 0$, this is referred to as pure (or pointwise) differential privacy and we may abbreviate $(\varepsilon, 0)$-DP to $\varepsilon$-DP. When $\delta > 0$, this is referred to as approximate differential privacy.*

The definition of pure DP was introduced by Dwork et al. (2006b) and approximate DP was introduced by Dwork et al. (2006a). Note that the notion of neighbouring datasets is context-dependent, but this is often glossed over. Our results are general and can be applied regardless of the specifics of what is a neighbouring dataset. (However, we do require symmetry – i.e., if $(x, x')$ are a pair of neighbouring inputs, then so are $(x'x)$.) Usually two datasets are said to be neighbouring if they differ only by the addition/removal or replacement of the data corresponding to a single individual. Some papers only consider addition or removal of a person's records, rather than replacement. But these are equivalent up to a factor of two.

In order to define Rényi DP, we first define the Rényi divergences:

**Definition 9** (Rényi Divergences). *Let $P$ and $Q$ be probability distributions on a common space $\Omega$. Assume that $P$ is absolutely continuous with respect to $Q$ – i.e., for all measurable $S \subset \Omega$, if $Q(S) = 0$, then $P(S) = 0$. Let $P(x)$ and $Q(x)$ denote the densities of $P$ and $Q$ respectively.[7] The KL divergence from $P$ to $Q$ is defined as*

$$\mathrm{D}_1\left(P\|Q\right) := \underset{X \leftarrow P}{\mathbb{E}}\left[\log\left(\frac{P(X)}{Q(X)}\right)\right] = \int_{\Omega} P(x) \log\left(\frac{P(x)}{Q(x)}\right) \mathrm{d}x.$$

*The max divergence from $P$ to $Q$ is defined as*

$$\mathrm{D}_{\infty}\left(P\|Q\right) := \sup\left\{\log\left(\frac{P(S)}{Q(S)}\right) : P(S) > 0\right\}.$$

*For $\lambda \in (1, \infty)$, the Rényi divergence from $P$ to $Q$ of order $\lambda$ is defined as*

$$\mathrm{D}_{\lambda}\left(P\|Q\right) := \frac{1}{\lambda - 1} \log\left(\underset{X \leftarrow P}{\mathbb{E}}\left[\left(\frac{P(X)}{Q(X)}\right)^{\lambda-1}\right]\right)$$

$$= \frac{1}{\lambda - 1} \log\left(\underset{X \leftarrow Q}{\mathbb{E}}\left[\left(\frac{P(X)}{Q(X)}\right)^{\lambda}\right]\right)$$

$$= \frac{1}{\lambda - 1} \log\left(\int_{\Omega} P(x)^{\lambda} Q(x)^{1-\lambda} \mathrm{d}x\right).$$

---

[7]In general, we can only define the ratio $P(x)/Q(x)$ to be the Radon-Nikodym derivative of $P$ with respect to $Q$. To talk about $P(x)$ and $Q(x)$ separately we must assume some base measure with respect to which these are defined. In most cases the base measure is either the counting measure in the case of discrete distributions or the Lesbesgue measure in the case of continuous distributions.

We state some basic properties of Rényi divergences; for further information see, e.g., Van Erven & Harremos (2014).

**Lemma 10.** *Let $P$, $Q$, $P'$, and $Q'$ be probability distributions. Let $\lambda \in [1, \infty]$. The following hold.*

- ***Non-negativity:*** $D_\lambda (P\|Q) \geq 0$.

- ***Monotonicity & Continuity:*** $D_\lambda (P\|Q)$ *is a continuous and non-decreasing function of $\lambda$.*

- ***Data processing inequality (a.k.a. Postprocessing):*** *Let $f(P)$ denote the distribution obtained by applying some (possibly randomized) function to a sample from $P$ and let $f(Q)$ denote the distribution obtained by applying the same function to a sample from $Q$. Then $D_\lambda (f(P)\|f(Q)) \leq D_\lambda (P\|Q)$.*

- ***Finite case suffices:*** *We have $D_\lambda (P\|Q) = \sup_f D_\lambda (f(P)\|f(Q))$ even when $f$ is restricted to functions with a finite range.*

- ***Chain rule (a.k.a. Composition):*** $D_\lambda (P \times P'\|Q \times Q') = D_\lambda (P\|Q) + D_\lambda (P'\|Q')$, *where $P \times P'$ and $Q \times Q'$ denote the product distributions of the individual distributions.*

- ***Convexity:*** *The function $(P, Q) \mapsto e^{(\lambda-1)D_\lambda(P\|Q)}$ is convex for all $\lambda \in (1, \infty)$. The function $(P, Q) \mapsto D_\lambda (P\|Q)$ convex if and only if $\lambda = 1$.*

Now we can state the definition of Rényi DP (RDP), which is due to Mironov (2017).

**Definition 11** (Rényi Differential Privacy). *A randomized algorithm $M : \mathcal{X}^n \to \mathcal{Y}$ is $(\lambda, \varepsilon)$-Rényi differentially private if, for all neighbouring pairs of inputs $x, x' \in \mathcal{X}^n$, $D_\lambda (M(x)\|M(x')) \leq \varepsilon$.*

A closely related definition is that of zero-concentrated differential privacy (Bun & Steinke, 2016) (which is based on an earlier definition (Dwork & Rothblum, 2016) of concentrated differential privacy that does not refer to Rényi divergences).

**Definition 12** (Concentrated Differential Privacy). *A randomized algorithm $M : \mathcal{X}^n \to \mathcal{Y}$ is $\rho$-zCDP if, for all neighbouring pairs of inputs $x, x' \in \mathcal{X}^n$ and all $\lambda \in (1, \infty)$, $D_\lambda (M(x)\|M(x')) \leq \rho \cdot \lambda$.*

Usually, we consider a family of $(\lambda, \varepsilon(\lambda))$-RDP guarantees, where $\varepsilon(\lambda)$ is a function, rather than a single function. Concentrated DP is one example of such a family, where the function is linear, and this captures the behaviour of many natural algorithms. In particular, adding Gaussian noise to a bounded sensitivity function: If $f : \mathcal{X}^n \to \mathbb{R}^d$ has sensitivity $\Delta$ – i.e., $\|f(x) - f(x')\|_2 \leq \Delta$ for all neighbouring $x, x'$ – and $M : \mathcal{X}^n \to \mathbb{R}^d$ is the algorithm that returns a sample from $\mathcal{N}(f(x), \sigma^2 I)$, then $M$ satisfies $\frac{\Delta^2}{2\sigma^2}$-zCDP.

We can convert from pure DP to concentrated or Rényi DP as follows (Bun & Steinke, 2016).

**Lemma 13.** *If $M$ satisfies $(\varepsilon, 0)$-differential privacy, then $M$ satisfies $\frac{1}{2}\varepsilon^2$-zCDP – i.e., $(\lambda, \frac{1}{2}\varepsilon^2\lambda)$-RDP for all $\lambda \in (1, \infty)$.*

Conversely, we can convert from concentrated or Rényi DP to approximate DP as follows (Canonne et al., 2020).

**Lemma 14.** *If $M$ satisfies $(\lambda, \hat{\varepsilon})$-RDP, then $M$ satisfies $(\varepsilon, \delta)$-DP where $\varepsilon \geq 0$ is arbitrary and*

$$\delta = \frac{\exp((\lambda - 1)(\hat{\varepsilon} - \varepsilon))}{\lambda} \cdot \left(1 - \frac{1}{\lambda}\right)^{\lambda-1}.$$

### A.2 Probability Generating Functions

Let $K$ be a random variable supported on $\mathbb{N} \cup \{0\}$. The probability generating function (PGF) of $K$ is defined by

$$f(x) = \mathbb{E}\left[x^K\right] = \sum_{k=0}^{\infty} \mathbb{P}\left[K = k\right] \cdot x^k.$$

The PGF $f(x)$ is always defined for $x \in [0, 1]$, but may or may not be defined for $x > 1$. The PGF characterizes $K$. In particular, we can recover the probability mass function from the derivatives of

the PGF (hence the name):

$$\mathbb{P}\left[K = k\right] = \frac{f^{(k)}(0)}{k!},$$

where $f^{(k)}(x)$ denotes the $k^{\text{th}}$ derivative of $f(x)$ and, in particular, $\mathbb{P}\left[K = 0\right] = f(0)$. We remark that is is often easiest to specify the PGF and derive the probability distribution from it, rather than vice versa; indeed, we arrived at the truncated negative binomial distribution by starting with the PGF that we want and then differentiating.

We can also easily recover the moments of $K$ from the PGF: We have $f^{(k)}(x) = \sum_{\ell=k}^{\infty} \mathbb{P}\left[K = \ell\right] \cdot x^{\ell-k} \cdot \ell \cdot (\ell - 1) \cdot (\ell - 2) \cdots (\ell - k + 1)$. In particular, $f(1) = \mathbb{E}\left[1\right] = 1$ and $f'(1) = \mathbb{E}\left[K\right]$ and $f''(1) = \mathbb{E}\left[K(K - 1)\right]$. Note that the PGF is a rescaling of the moment generating function (MGF) $g(t) := \mathbb{E}\left[e^{tK}\right] = f(e^t)$.

The PGF can be related to the MGF in another way: Suppose $\Lambda$ is a random variable on $[0, \infty)$. Now suppose we draw $K \leftarrow \mathsf{Poisson}(\Lambda)$. Then the PGF of $K$ is the MGF of $\Lambda$ – i.e., $\mathbb{E}\left[x^K\right] = \mathbb{E}_{\Lambda}\left[\mathbb{E}_{K \leftarrow \mathsf{Poisson}(\Lambda)}\left[x^K\right]\right] = \mathbb{E}_{\Lambda}\left[e^{\Lambda \cdot (x-1)}\right] = g(x - 1)$, where $g$ is the MGF of $\Lambda$. In particular, if $\Lambda$ is drawn from a Gamma distribution, then this would yield $K$ from a negative binomial distribution which has a PGF of the form $f_{\text{NB}}(x) = \left(\frac{1-(1-\gamma)x}{\gamma}\right)^{-\eta}$.[8] Note that our results work with a truncated negative binomial distribution, which is a negative binomial conditioned on $K \neq 0$. This corresponds to an affine rescaling of the PGF, namely $f_{\text{TNB}}(x) = \frac{f_{\text{NB}}(x) - f_{\text{NB}}(0)}{f_{\text{NB}}(1) - f_{\text{NB}}(0)}$.

We can also obtain a negative binomial distribution as a compound of a Poisson distribution and a logarithmic distribution. That is, if we draw $T$ from a Poisson distribution and draw $K_1, K_2, \cdots, K_T$ independently from a logarithmic distribution, then $K = \sum_{t=1}^{T} K_t$ follows a negative binomial distribution. The PGF of the logarithmic distribution is given by $f_{K_t}(x) = \mathbb{E}\left[x^{K_t}\right] = \frac{\log(1-(1-\gamma)x)}{\log(\gamma)}$ and the PGF of Poisson is given by $f_T(x) = \mathbb{E}\left[x^T\right] = e^{\mu \cdot (x-1)}$. Hence

$$f_K(x) = \mathbb{E}\left[x^K\right] = \mathbb{E}_T\left[\prod_{t=1}^{T} \mathbb{E}_{K_t}\left[x^{K_t}\right]\right] = \mathbb{E}_T\left[f_{K_t}(x)^T\right] = f_T(f_{K_t}(x)) = \exp\left(\mu \cdot \left(\frac{\log(1-(1-\gamma)x)}{\log(\gamma)} - 1\right)\right),$$

which is equivalent to $f_{\text{NB}}(x) = \left(\frac{1-(1-\gamma)x}{\gamma}\right)^{-\eta}$ with $\eta = \frac{\mu}{\log(1/\gamma)}$.

Finally we remark that we can also use the PGF to show convergence in probability. In particular,

$$\lim_{\eta \to \infty, \gamma = \frac{\eta}{\eta+\mu}} f_{\text{NB}}(x) = \lim_{\eta \to \infty, \gamma = \frac{\eta}{\eta+\mu}} \left(\frac{1 - (1 - \gamma)x}{\gamma}\right)^{-\eta} = \lim_{\eta \to \infty, \gamma = \frac{\eta}{\eta+\mu}} \left(1 - \frac{\mu}{\eta}(x - 1)\right)^{-\eta} = e^{\mu(x-1)}.$$

That is, if we take the limit of the negative binomial distribution as $\eta \to \infty$ but the mean $\mu = \eta\frac{1-\gamma}{\gamma}$ remains fixed, then we obtain a Poisson distribution. If we take $\eta \to 0$, then $f_{\text{NB}}(x) \to 1$, which is to say that the negative binomial distribution converges to a point mass at 0 as $\eta \to 0$. However, the *truncated* negative binomial distribution converges to a logarithmic distribution as $\eta \to 0$.

### A.2.1 PROBABILITY GENERATING FUNCTIONS AND UTILITY

Recall that in Section 3.6, we analyzed the expected utility and runtime of different distributions on the number of repetitions $K$. Given our discussion of probability generating functions for these distributions, we can offer an alternative perspective on the expected utility and runtime.

Suppose each invocation of $Q$ has a probability $1/m$ of producing a "good" output. This would be the case if we are considering $m$ hyperparameter settings and only one is good—where here we consider the outcome to be binary (good or bad) for simplicity and what is a good or bad is determined only by the total order on the range $\mathcal{Y}$ and some threshold on the quality score (e.g., accuracy). Then $A$ has a probability

$$\beta := 1 - \mathbb{P}\left[A(x) \in \text{Bad}\right] = 1 - \mathbb{E}_K\left[\mathbb{P}\left[Q(x) \in \text{Bad}\right]^K\right] = 1 - \mathbb{E}\left[(1 - 1/m)^K\right] = 1 - f(1 - 1/m)$$

---

[8]The PGF of $\mathsf{Binomial}(n, p)$ is $f(x) = (1 - p + px)^n$. This expression is similar to the PGF of the negative binomial, except the negative binomial has a negative exponent.

of outputting a good output, where $f(x) = \mathbb{E}\left[x^K\right]$ is the probability generating function of the distribution. If we make the first-order approximation $f(1 - 1/m) \approx f(1) - f'(1) \cdot 1/m = 1 - \mathbb{E}\left[K\right]/m$, then we have $\beta \approx \mathbb{E}\left[K\right]/m$. In other words, for small values of $1/m$, the probability of success is amplified by a multiplicative factor of $\mathbb{E}\left[K\right]$.

However, the above first-order approximation only holds for large $m$ and, hence, small overall success probabilities $\beta$. In practice, we want $\beta \approx 1$. The different distributions (Poisson and truncated negative binomial with different values of $\eta$) have very different behaviours even with the same expectation. In the regime where we want the overall success probability to be high (i.e., $\beta \approx 1$), smaller $\eta$ performs worse, because the distribution is more heavy-tailed. The best performing distribution is the Poisson distribution, which is almost as concentrated as naïve repetition. Figure 6 show the success probability $\beta$ as a function of the final $(\varepsilon, 10^{-6})$-DP guarantee. This demonstrates that there is a tradeoff between distributions.

More generally, we can relate the PGF of $K$ to the expected utility of our repeated algorithm. Let $X \in \mathbb{R}$ be random variable corresponding to the utility of one run of the base algorithm $Q$. E.g. $X$ could represent the accuracy, loss, AUC/AUROC, or simply the quantile of output. Now let $Y \in \mathbb{R}$ be the utility of our repeated algorithm $A$ which runs the base algorithm $Q$ repeatedly $K$ times for a random $K$. That is, $Y = \max\{X_1, \cdots, X_K\}$ where $X_1, X_2, \cdots$ are independent copies of $X$. Let $\mathrm{cdf}_X(x) = \mathbb{P}\left[X \leq x\right]$ and

$$\mathrm{cdf}_Y(x) = \mathbb{P}\left[Y \leq x\right] = \mathbb{E}_K\left[\mathbb{P}_X\left[X \leq x\right]^K\right] = f(\mathrm{cdf}_X(x)),$$

where $f(x) = \mathbb{E}\left[x^K\right]$ is the PGF of the number of repetitions $K$. Assuming for the moment that $X$ is a continuous random variable, we can derive the probability density function of $Y$ from the cumulative distribution function:

$$\mathrm{pdf}_Y(x) = \frac{\mathrm{d}}{\mathrm{d}x}\mathrm{cdf}_X(x) = \frac{\mathrm{d}}{\mathrm{d}x}f(\mathrm{cdf}_X(x)) = f'(\mathrm{cdf}_X(x)) \cdot \mathrm{pdf}_X(x).$$

This allows us to compute the expected utility:

$$\mathbb{E}\left[Y\right] = \int_{-\infty}^{\infty} x \cdot \mathrm{pdf}_Y(x)\mathrm{d}x = \int_{-\infty}^{\infty} x \cdot f'(\mathrm{cdf}_X(x)) \cdot \mathrm{pdf}_X(x)\mathrm{d}x = \mathbb{E}\left[X \cdot f'(\mathrm{cdf}_X(X))\right].$$

In particular, we can compute the expected quantile (8) in which case $X$ is uniform on $[0, 1]$ and, hence, $\mathrm{cdf}_X(x) = x$ and $\mathrm{pdf}_X(x) = 1$ for $x \in [0, 1]$. Integration by parts gives

$$\mathbb{E}\left[Y\right] = \int_0^1 x \cdot f'(x)\mathrm{d}x = \int_0^1 \left(\frac{\mathrm{d}}{\mathrm{d}x}xf(x)\right) - f(x)\mathrm{d}x = 1f(1) - 0f(0) - \int_0^1 f(x)\mathrm{d}x = 1 - \int_0^1 f(x)\mathrm{d}x.$$

Note that

$$\int_0^1 f(x)\mathrm{d}x = \int_0^1 \mathbb{E}_K\left[x^K\right]\mathrm{d}x = \mathbb{E}_K\left[\int_0^1 x^K\mathrm{d}x\right] = \mathbb{E}_K\left[\frac{1}{K+1}\right].$$

Finally, we also want to ensure that the runtime of our hyperparameter tuning algorithm is well-behaved. In particular, we wish to avoid heavy-tailed runtimes. We can obtain tail bounds on the number of repetitions $K$ from the PGF or MGF too: For all $t > 0$, we have

$$\mathbb{P}\left[K \geq k\right] = \mathbb{P}\left[e^{t \cdot (K-k)} \geq 1\right] \leq \mathbb{E}\left[e^{t \cdot (K-k)}\right] = f(e^t) \cdot e^{-t \cdot k}.$$

Thus, if the PGF $f(x) = \mathbb{E}\left[x^K\right]$ is finite for some $x = e^t > 1$, then we obtain a subexponential tail bound on $K$.

# B    PROOFS FROM SECTION 3

## B.1    PROOF OF GENERIC BOUND

*Proof of Lemma 7.* We assume that $\mathcal{Y}$ is a finite set and that $\mathbb{P}\left[K = 0\right] = 0$; this is, essentially, without loss of generality.[9] Denote $Q(\leq y) := \sum_{y' \in \mathcal{Y}} \mathbb{I}[y' \leq y] \cdot Q(y')$ and similarly for $Q(< y)$

---

[9]Our proof can be extended to the general case. Alternatively, if $\mathcal{Y}$ is infinite, we can approximate the relevant quantities arbitrarily well with a finite partition; see Lemma 10 or Van Erven & Harremos (2014, Theorem 10).

and analogously for $Q'$ in place of $Q$. For each $y \in \mathcal{Y}$, we have

$$
\begin{aligned}
A(y) &= \sum_{k=1}^{\infty} \mathbb{P}\left[K = k\right] \cdot (Q(\leq y)^k - Q(< y)^k) \\
&= f(Q(\leq y)) - f(Q(< y)) \\
&= \int_{Q(<y)}^{Q(\leq y)} f'(x)\mathrm{d}x \\
&= Q(y) \cdot \underset{\substack{X \leftarrow [Q(<y), Q(\leq y)] \\ \text{uniform}}}{\mathbb{E}} [f'(X)]
\end{aligned}
$$

and, likewise, $A'(y) = Q'(y) \cdot \underset{\substack{X' \leftarrow [Q'(<y), Q'(\leq y)] \\ \text{uniform}}}{\mathbb{E}} [f'(X')]$. Thus

$$
\begin{aligned}
e^{(\lambda-1)\mathrm{D}_\lambda\left(A \| A'\right)} &= \sum_{y \in \mathcal{Y}} A(y)^\lambda \cdot A'(y)^{1-\lambda} \\
&= \sum_{y \in \mathcal{Y}} Q(y)^\lambda \cdot Q'(y)^{1-\lambda} \cdot \underset{X \leftarrow [Q(<y), Q(\leq y)]}{\mathbb{E}} [f'(X)]^\lambda \cdot \underset{X' \leftarrow [Q'(<y), Q'(\leq y)]}{\mathbb{E}} [f'(X')]^{1-\lambda} \\
&\leq \sum_{y \in \mathcal{Y}} Q(y)^\lambda \cdot Q'(y)^{1-\lambda} \cdot \underset{\substack{X \leftarrow [Q(<y), Q(\leq y)] \\ X' \leftarrow [Q'(<y), Q'(\leq y)]}}{\mathbb{E}} \left[f'(X)^\lambda \cdot f'(X')^{1-\lambda}\right] \\
&\leq e^{(\lambda-1)\mathrm{D}_\lambda\left(Q \| Q'\right)} \cdot \max_{y \in \mathcal{Y}} \underset{\substack{X \leftarrow [Q(<y), Q(\leq y)] \\ X' \leftarrow [Q'(<y), Q'(\leq y)]}}{\mathbb{E}} \left[f'(X)^\lambda \cdot f'(X')^{1-\lambda}\right].
\end{aligned}
$$

The second inequality follows from Hölder's inequality. The first inequality follows from the fact that, for any $\lambda \in \mathbb{R}$, the function $h : (0, \infty)^2 \to (0, \infty)$ given by $h(u, v) = u^\lambda \cdot v^{1-\lambda}$ is convex and, hence, $\mathbb{E}\left[U\right]^\lambda \mathbb{E}\left[V\right]^{1-\lambda} = h(\mathbb{E}\left[(U, V)\right]) \leq \mathbb{E}\left[h(U, V)\right] = \mathbb{E}\left[U^\lambda \cdot V^{1-\lambda}\right]$ for any pair of positive random variables $(U, V)$. Note that we require $X$ to be uniform on $[Q(< y), Q(\leq y)]$ and $X'$ to be uniform on $[Q'(< y), Q'(\leq y)]$, but their joint distribution can be arbitrary. We will couple them so that $\frac{X - Q(<y)}{Q(y)} = \frac{X' - Q'(<y)}{Q'(y)}$. In particular, this implies that, for each $y \in \mathcal{Y}$, there exists some $t \in [0, 1]$ such that

$$
\underset{\substack{X \leftarrow [Q(<y), Q(\leq y)] \\ X' \leftarrow [Q'(<y), Q'(\leq y)]}}{\mathbb{E}} \left[f'(X)^\lambda \cdot f'(X')^{1-\lambda}\right] \leq f'(Q(< y) + t \cdot Q(y))^\lambda \cdot f'(Q'(< y) + t \cdot Q'(y))^{1-\lambda}.
$$

Hence

$$
\mathrm{D}_\lambda\left(A \| A'\right) \leq \mathrm{D}_\lambda\left(Q \| Q'\right) + \frac{1}{\lambda - 1} \log \left( \max_{\substack{y \in \mathcal{Y} \\ t \in [0,1]}} f'(Q(< y) + t \cdot Q(y))^\lambda \cdot f'(Q'(< y) + t \cdot Q'(y))^{1-\lambda} \right).
$$

To prove the result, we simply fix $y_* \in \mathcal{Y}$ and $t_* \in [0, 1]$ achieving the maximum above and define

$$
g(y) := \begin{cases} 1 & \text{if } y < y_* \\ t_* & \text{if } y = y_* \\ 0 & \text{if } y > y_* \end{cases}.
$$

$\square$

## B.2 PROOFS OF DISTRIBUTION-SPECIFIC BOUNDS

**Truncated Negative Binomial Distribution**

*Proof of Theorem 2.* The probability generating function of the truncated negative binomial distribution is

$$
f(x) = \underset{K \leftarrow \mathcal{D}_{\eta,\gamma}}{\mathbb{E}} \left[x^K\right] = \begin{cases} \frac{(1-(1-\gamma)x)^{-\eta}-1}{\gamma^{-\eta}-1} & \text{if } \eta \neq 0 \\ \frac{\log(1-(1-\gamma)x)}{\log(\gamma)} & \text{if } \eta = 0 \end{cases}.
$$

Thus

$$f'(x) = (1 - (1-\gamma)x)^{-\eta-1} \cdot \begin{cases} \frac{\eta \cdot (1-\gamma)}{\gamma^{-\eta}-1} & \text{if } \eta \neq 0 \\ \frac{1-\gamma}{\log(1/\gamma)} & \text{if } \eta = 0 \end{cases}$$
$$= (1 - (1-\gamma)x)^{-\eta-1} \cdot \gamma^{\eta+1} \cdot \mathbb{E}[K].$$

Now we delve into the privacy analysis: Let $Q = Q(x)$ and $Q' = Q(x')$ denote the output distributions of $Q$ on two neighbouring inputs. Similarly, let $A = A(x)$ and $A' = A(x')$ be the corresponding pair of output distributions of the repeated algorithm. By Lemma 7, for appropriate values $q, q' \in [0,1]$ and for all $\lambda > 1$ and all $\hat{\lambda} > 1$,[10] we have

$D_\lambda(A\|A')$

$\leq D_\lambda(Q\|Q') + \frac{1}{\lambda-1} \log \left( f'(q)^\lambda \cdot f'(q')^{1-\lambda} \right)$

$= D_\lambda(Q\|Q') + \frac{1}{\lambda-1} \log \left( \gamma^{\eta+1} \cdot \mathbb{E}[K] \cdot (1-(1-\gamma)q)^{-\lambda(\eta+1)} \cdot (1-(1-\gamma)q')^{-(1-\lambda)(\eta+1)} \right)$

$= D_\lambda(Q\|Q') + \frac{1}{\lambda-1} \log \left( \gamma^{\eta+1} \cdot \mathbb{E}[K] \cdot \left( (\gamma + (1-\gamma)(1-q))^{1-\hat{\lambda}} \cdot (\gamma + (1-\gamma)(1-q'))^{\hat{\lambda}} \right)^\nu \cdot (\gamma + (1-\gamma)(1-q))^u \right)$

$\qquad\qquad (\hat{\lambda}\nu = (\lambda-1)(1+\eta) \text{ and } (1-\hat{\lambda})\nu + u = -\lambda(\eta+1))$

$\leq D_\lambda(Q\|Q') + \frac{1}{\lambda-1} \log \left( \gamma^{\eta+1} \cdot \mathbb{E}[K] \cdot \left( \gamma + (1-\gamma) \cdot e^{(\hat{\lambda}-1)D_{\hat{\lambda}}(Q'\|Q)} \right)^\nu \cdot (\gamma + (1-\gamma)(1-q))^u \right)$

$(1-q \text{ and } 1-q' \text{ are postprocessings of } Q \text{ and } Q' \text{ respectively and } e^{(\hat{\lambda}-1)D_{\hat{\lambda}}(\cdot\|\cdot)} \text{ is convex and } \nu \geq 0)$

$\leq D_\lambda(Q\|Q') + \frac{1}{\lambda-1} \log \left( \gamma^{\eta+1} \cdot \mathbb{E}[K] \cdot \left( \gamma + (1-\gamma) \cdot e^{(\hat{\lambda}-1)D_{\hat{\lambda}}(Q'\|Q)} \right)^\nu \cdot \gamma^u \right)$

$\qquad\qquad (\gamma \leq \gamma + (1-\gamma)(1-q) \text{ and } u \leq 0)$

$= D_\lambda(Q\|Q') + \frac{\nu}{\lambda-1} \log \left( \gamma + (1-\gamma) \cdot e^{(\hat{\lambda}-1)D_{\hat{\lambda}}(Q'\|Q)} \right) + \frac{1}{\lambda-1} \log \left( \gamma^{\eta+1} \cdot \mathbb{E}[K] \cdot \gamma^u \right)$

$= D_\lambda(Q\|Q') + \frac{\nu}{\lambda-1} \left( (\hat{\lambda}-1)D_{\hat{\lambda}}(Q'\|Q) + \log \left( 1 - \gamma \cdot \left( 1 - e^{-(\hat{\lambda}-1)D_{\hat{\lambda}}(Q'\|Q)} \right) \right) \right)$

$\quad + \frac{1}{\lambda-1} \log \left( \gamma^{u+\eta+1} \cdot \mathbb{E}[K] \right)$

$= D_\lambda(Q\|Q') + (1+\eta) \left( 1 - \frac{1}{\hat{\lambda}} \right) D_{\hat{\lambda}}(Q'\|Q) + \frac{1+\eta}{\hat{\lambda}} \log \left( 1 - \gamma \cdot \left( 1 - e^{-(\hat{\lambda}-1)D_{\hat{\lambda}}(Q'\|Q)} \right) \right)$

$\quad + \frac{\log(\mathbb{E}[K])}{\lambda-1} + \frac{1+\eta}{\hat{\lambda}} \log(1/\gamma) \qquad (\nu = \frac{(\lambda-1)(1+\eta)}{\hat{\lambda}} \text{ and } u = -(1+\eta)(\frac{\lambda-1}{\hat{\lambda}}+1))$

$= D_\lambda(Q\|Q') + (1+\eta) \left( 1 - \frac{1}{\hat{\lambda}} \right) D_{\hat{\lambda}}(Q'\|Q) + \frac{1+\eta}{\hat{\lambda}} \log \left( \frac{1}{\gamma} - 1 + e^{-(\hat{\lambda}-1)D_{\hat{\lambda}}(Q'\|Q)} \right) + \frac{\log(\mathbb{E}[K])}{\lambda-1}$

$\leq D_\lambda(Q\|Q') + (1+\eta) \left( 1 - \frac{1}{\hat{\lambda}} \right) D_{\hat{\lambda}}(Q'\|Q) + \frac{1+\eta}{\hat{\lambda}} \log \left( \frac{1}{\gamma} \right) + \frac{\log(\mathbb{E}[K])}{\lambda-1}.$

$\qquad\qquad\qquad\qquad\qquad\qquad\qquad\qquad\qquad\qquad\qquad\qquad\qquad\qquad\qquad\qquad\qquad\square$

*Proof of Corollary 4.* We assume that $Q : \mathcal{X}^n \to \mathcal{Y}$ is a randomized algorithm satisfying $\rho$-zCDP – i.e., $(\lambda, \rho \cdot \lambda)$-Rényi DP for all $\lambda > 1$. Substituting this guarantee into Theorem 2 (i.e., setting $\varepsilon = \rho \cdot \lambda$ and $\hat{\varepsilon} = \rho \cdot \hat{\lambda}$) gives that the repeated algorithm $A$ satisfies $(\lambda, \varepsilon')$-RDP for

$$\varepsilon' \leq \rho \cdot \lambda + (1+\eta) \cdot \left( 1 - \frac{1}{\hat{\lambda}} \right) \rho \cdot \hat{\lambda} + \frac{(1+\eta) \cdot \log(1/\gamma)}{\hat{\lambda}} + \frac{\log \mathbb{E}[K]}{\lambda-1}.$$

This holds for all $\lambda \in (1, \infty)$ and all $\hat{\lambda} \in [1, \infty)$.

---

[10]Our proof here assumes $\hat{\lambda} > 1$, but the result holds for $\hat{\lambda} = 1$ too. This can be shown either by analyzing this case separately or simply by taking the limit $\hat{\lambda} \to 1$ and using the continuity properties of Rényi divergence.

We set $\hat{\lambda} = \sqrt{\log(1/\gamma)/\rho}$ to minimize this expression. Note that we assume $\rho \le \log(1/\gamma)$ and hence this is a valid setting of $\hat{\lambda} \ge 1$. This reduces the expression to

$$\varepsilon' \le \rho \cdot \lambda - (1 + \eta) \cdot \rho + 2(1 + \eta) \cdot \sqrt{\rho \cdot \log(1/\gamma)} + \frac{\log \mathbb{E}\,[K]}{\lambda - 1}.$$

This bound is minimized when $\lambda - 1 = \sqrt{\log(\mathbb{E}\,[K])/\rho}$. If $\lambda - 1 < \sqrt{\log(\mathbb{E}\,[K])/\rho}$, then we can apply the monotonicity property of Renyi DP (Remark 5 and Lemma 10) and substitute in the bound with this optimal $\lambda$. That is, we obtain the bound

$$\varepsilon' \le \begin{cases} \rho \cdot \lambda - (1 + \eta) \cdot \rho + 2(1 + \eta) \cdot \sqrt{\rho \cdot \log(1/\gamma)} + \frac{\log \mathbb{E}[K]}{\lambda - 1} & \text{if } \lambda > 1 + \sqrt{\log(\mathbb{E}\,[K])/\rho} \\ 2\sqrt{\rho \cdot \log \mathbb{E}\,[K]} + 2(1 + \eta)\sqrt{\rho \cdot \log(1/\gamma)} - \eta\rho & \text{if } \lambda \le 1 + \sqrt{\log(\mathbb{E}\,[K])/\rho} \end{cases}.$$

$\square$

**Proof of the Poisson Distribution Bound**

*Proof of Theorem 6.* The probability generating function for the Poisson distribution is $f(x) = \mathbb{E}\,[x^K] = e^{\mu(x-1)}$. Thus $f'(x) = \mu \cdot e^{\mu(x-1)}$. As in the previous proofs, let $x$ and $x'$ be neighbouring inputs. Denote $Q = Q(x)$, $Q' = Q(x')$, $A = A(x)$, and $A' = A(x)$. By Lemma 7,

$$\mathrm{D}_\lambda\,(A\|A')$$

$$\le \mathrm{D}_\lambda\,(Q\|Q') + \frac{1}{\lambda - 1} \log \left( f'(q)^\lambda \cdot f'(q')^{1-\lambda} \right)$$

$$= \mathrm{D}_\lambda\,(Q\|Q') + \frac{1}{\lambda - 1} \log \left( \mu \cdot e^{\mu\lambda(q-1)+\mu(1-\lambda)(q'-1)} \right)$$

$$= \mathrm{D}_\lambda\,(Q\|Q') + \frac{\mu(\lambda q - (\lambda - 1)q' - 1) + \log \mu}{\lambda - 1}$$

$$= \mathrm{D}_\lambda\,(Q\|Q') + \frac{\mu((\lambda - 1)(1 - q') - \lambda(1 - q)) + \log \mu}{\lambda - 1}$$

$$\le \mathrm{D}_\lambda\,(Q\|Q') + \frac{\mu((\lambda - 1)(e^{\hat{\varepsilon}}(1 - q) + \hat{\delta}) - \lambda(1 - q)) + \log \mu}{\lambda - 1}$$

(by our $(\hat{\varepsilon}, \hat{\delta})$-DP assumption on $Q$ and since $1 - q$ and $1 - q'$ are postprocessings)

$$= \mathrm{D}_\lambda\,(Q\|Q') + \mu \cdot (1 - q) \cdot \left( e^{\hat{\varepsilon}} - \frac{\lambda}{\lambda - 1} \right) + \mu \cdot \hat{\delta} + \frac{\log \mu}{\lambda - 1}$$

$$\le \mathrm{D}_\lambda\,(Q\|Q') + \mu \cdot \hat{\delta} + \frac{\log \mu}{\lambda - 1},$$

where the final inequality follows from our assumption that $e^{\hat{\varepsilon}} \le 1 + \frac{1}{\lambda - 1}$.

$\square$

## C  CONDITIONAL SAMPLING APPROACH

In the main text, we analyzed the approach where we run the underlying algorithm $Q$ a random number of times according to a carefully-chosen distribution and output the best result from these independent runs. An alternative approach – also studied by Liu & Talwar (2019) – is to start with a pre-defined threshold for a "good enough" output and to run $Q$ repeatedly until it produces such a result and then output that. This approach has some advantages, namely being simpler and avoiding the heavy-tailed behaviour of the logarithmic distribution while attaining the same kind of privacy guarantee. However, the disadvantage of this approach is that we must specify the acceptance threshold a priori. If we set the threshold too high, then we may have to keep running $Q$ for a long time.[11] If we set the threshold too low, then we may end up with a suboptimal output.

---

[11]One solution to avoid an overly long runtime in the case that $Q(S)$ is too small is to modify $Q$ to output $\perp$ with some small probability $p$ and then have $A$ halt if this occurs. This would represent a failure of the algorithm to produce a good output, but would avoid a privacy failure.

We analyze this approach under Rényi DP, thereby extending the results of Liu & Talwar (2019). Our algorithm $A$ now works as follows. We start with a base algorithm $Q$ and a set of good outputs $S$. Now $A(x)$ computes $y = Q(x)$ and, if $y \in S$, then it returns $y$ and halts. Otherwise, $A$ repeats the procedure. This is equivalent to sampling from a conditional distribution $Q(x)|Q(x) \in S$. The number of times $Q$ is run will follow a geometric distribution with mean $1/Q(S)$.

**Proposition 15.** *Let $\lambda \in (1, \infty)$. Let $Q$ and $Q'$ be probability distributions on $\Omega$ with $\mathrm{D}_\lambda(Q\|Q') < \infty$. Let $S \subset \Omega$ have nonzero measure under $Q'$ and also under $Q$. Let $Q_S$ and $Q'_S$ denote the conditional distributions of $Q$ and $Q'$ respectively conditioned on being in the set $S$. That is, $Q_S(E) = Q(E \cap S)/Q(S)$ and $Q'_S(E) = Q'(E \cap S)/Q'(S)$ for all measurable $E \subset \Omega$. Then, for all $p, q, r \in [1, \infty]$ satisfying $1/p + 1/q + 1/r = 1$, we have*

$$\mathrm{D}_\lambda(Q_S\|Q'_S) \leq \frac{\lambda - 1/p - 1/r}{\lambda - 1}\mathrm{D}_{r \cdot (\lambda - 1/p)}(Q\|Q') + \frac{\lambda + 1/q - 2}{\lambda - 1}\mathrm{D}_{\lambda + 1/q - 1}(Q'\|Q) + \frac{1/r + 1}{\lambda - 1}\log\left(\frac{1}{Q(S)}\right).$$

*Proof.* For $x \in \Omega$, denote the various distributional densities at $x$ (relative to some base measure) by $Q_S(x)$, $Q'_S(x)$, $Q(x)$, and $Q'(x)$. We have $Q_S(x) = Q(x)\mathbb{I}[x \in S]/Q(S)$ and $Q'_S(x) = Q'(x)\mathbb{I}[x \in S]/Q'(S)$. Now we have

$$e^{(\lambda - 1)\mathrm{D}_\lambda\left(Q_S\|Q'_S\right)} = \int_\Omega Q_S(x)^\lambda Q'_S(x)^{1-\lambda}\mathrm{d}x$$

$$= Q(S)^{-\lambda}Q'(S)^{\lambda - 1}\int_\Omega \mathbb{I}[x \in S]Q(x)^\lambda Q'(x)^{1-\lambda}\mathrm{d}x$$

$$\leq Q(S)^{-\lambda}Q'(S)^{\lambda - 1}\left(\int_S Q(x)\mathrm{d}x\right)^{1/p}\left(\int_S Q'(x)\mathrm{d}x\right)^{1/q}\left(\int_S \left(Q(x)^{\lambda - 1/p}Q'(x)^{1-\lambda - 1/q}\right)^r \mathrm{d}x\right)^{1/r}$$

(Hölder's inequality)

$$= Q(S)^{1/p - \lambda}Q'(S)^{1/q + \lambda - 1}\left(\int_S Q(x)^{r\lambda - r/p}Q'(x)^{r - r\lambda - r/q}\mathrm{d}x\right)^{1/r}$$

$$= Q'(S)^{\lambda_0}Q(S)^{1-\lambda_0}Q(S)^{-1/r - 1}\left(\int_S Q(x)^{\lambda_1}Q'(x)^{1-\lambda_1}\mathrm{d}x\right)^{1/r}$$

($\lambda_0 := \lambda + 1/q - 1$, $\lambda_1 := r\lambda - r/p$)

$$\leq e^{(\lambda_0 - 1)\mathrm{D}_{\lambda_0}\left(Q'\|Q\right)} \cdot Q(S)^{-1/r - 1} \cdot \left(e^{(\lambda_1 - 1)\mathrm{D}_{\lambda_1}\left(Q\|Q'\right)}\right)^{1/r}.$$

(Postprocessing & non-negativity)

□

The number of parameters in Proposition 15 is excessive. Thus we provide some corollaries that simplify the expression somewhat.

**Corollary 16.** *Let $\lambda, Q, Q', S, Q_S, Q'_S$ be as in Proposition 15. The following inequalities all hold.*

$$\mathrm{D}_\infty(Q_S\|Q'_S) \leq \mathrm{D}_\infty(Q\|Q') + \mathrm{D}_\infty(Q'\|Q).$$

$$\mathrm{D}_\lambda(Q_S\|Q'_S) \leq \mathrm{D}_\lambda(Q\|Q') + \frac{\lambda - 2}{\lambda - 1}\mathrm{D}_{\lambda - 1}(Q'\|Q) + \frac{2}{\lambda - 1}\log\left(\frac{1}{Q(S)}\right).$$

$$\mathrm{D}_\lambda(Q_S\|Q'_S) \leq \mathrm{D}_\infty(Q\|Q') + \frac{\lambda - 2}{\lambda - 1}\mathrm{D}_{\lambda - 1}(Q'\|Q) + \frac{1}{\lambda - 1}\log\left(\frac{1}{Q(S)}\right).$$

$$\mathrm{D}_\lambda(Q_S\|Q'_S) \leq \frac{\lambda}{\lambda - 1}\mathrm{D}_\infty(Q\|Q') + \mathrm{D}_\lambda(Q'\|Q) + \frac{1}{\lambda - 1}\log\left(\frac{1}{Q(S)}\right).$$

$$\forall r \geq 1 \quad \mathrm{D}_\lambda(Q_S\|Q'_S) \leq \mathrm{D}_{r(\lambda - 1) + 1}(Q\|Q') + \frac{\lambda - 2}{\lambda - 1}\mathrm{D}_{\lambda - 1}(Q'\|Q) + \frac{1/r + 1}{\lambda - 1}\log\left(\frac{1}{Q(S)}\right).$$

The first inequality in Corollary 16 is essentially the result given by Liu & Talwar (2019): If $Q$ satisfies $\varepsilon$-DP, then $A$ satisfies $2\varepsilon$-DP.

Figure 8 plots the guarantee of the second inequality in Corollary 16 when $\mathrm{D}_\lambda(Q\|Q') = 0.1\lambda$ and $\mathrm{D}_{\lambda - 1}(Q'\|Q) = 0.1(\lambda - 1)$.

## D NEGATIVE RESULTS ON IMPROVEMENTS TO OUR ANALYSIS

It is natural to wonder whether our results could be further improved. In this section, we give some examples that demonstrates that quantitatively there is little room for improvement.

### D.1 WHY A FIXED NUMBER OF REPETITIONS DOES NOT RESULT IN GOOD PRIVACY.

We first consider more closely the strawman approach discussed in Section 3.2: the base algorithm $Q$ is repeated a fixed number of times $k$ and we return the best output. This corresponds to picking $k$ from a point mass distribution. To understand why it performs so poorly from a privacy standpoint, we first apply our main result from Section 3.5 to the resulting point mass distribution.

**Point Mass:** Suppose $K$ is just a point mass – i.e., $\mathbb{P}\left[K = k\right] = 1$. So $A$ runs the algorithm $Q$ a deterministic number of times. Then the probability generating function (PGF) is $f(x) = x^k$ and its derivative is $f'(x) = k \cdot x^{k-1}$. Let $Q$ denote the base algorithm. We abuse notation and let $Q = Q(x)$ and $Q' = Q(x')$, where $x$ and $x'$ are neighbouring inputs. Similarly, let $A = A(x)$ and $A' = A(x')$ be the final output distributions obtained by running $Q$ and $Q'$ repeatedly $k$ times and returning the best result. We follow the same pattern of analysis that we applied to the other distributions in Theorems 2 and 6: Lemma 7 gives the bound

$$
\begin{aligned}
\mathrm{D}_\lambda\left(A\|A'\right) &\le \mathrm{D}_\lambda\left(Q\|Q'\right) + \frac{1}{\lambda - 1} \log\left(k \cdot \left(q^\lambda \cdot q'^{1-\lambda}\right)^{k-1}\right) \\
&\le \mathrm{D}_\lambda\left(Q\|Q'\right) + \frac{1}{\lambda - 1} \log\left(k \cdot \left(e^{(\lambda-1)\mathrm{D}_\lambda\left(\mathsf{Bern}(q)\|\mathsf{Bern}(q')\right)}\right)^{k-1}\right) \\
&\le \mathrm{D}_\lambda\left(Q\|Q'\right) + \frac{1}{\lambda - 1} \log\left(k \cdot \left(e^{(\lambda-1)\mathrm{D}_\lambda\left(Q\|Q'\right)}\right)^{k-1}\right) \\
&= k \cdot \mathrm{D}_\lambda\left(Q\|Q'\right) + \frac{\log k}{\lambda - 1},
\end{aligned}
$$

where the final inequality follows from the fact that $\mathsf{Bern}(q)$ and $\mathsf{Bern}(q')$ are postprocessings of $Q$ and $Q'$ respectively.

This bound is terrible. In fact, it is slightly *worse* than a naïve composition analysis which would give $\mathrm{D}_\lambda\left(A\|A'\right) \le \mathrm{D}_\lambda\left(Q^{\otimes k}\|Q'^{\otimes k}\right) = k \cdot \mathrm{D}_\lambda\left(Q\|Q'\right)$. It shows that a deterministic number of repetitions does not yield good privacy parameters, at least with this analysis.

It is surprising that running the base algorithm $Q$ a fixed number of times $k$ and returning the best output performs so poorly from a privacy standpoint. We will now give a simple example that demonstrates that this is inherent and not just a limitation of our analysis. Liu & Talwar (2019, Appendix B) give a similar example.

**Proposition 17.** *For all $\varepsilon > 0$, there exists a $\varepsilon$-DP algorithm $Q : \mathcal{X}^n \to \mathcal{Y}$ such that the following holds. Define an algorithm $A : \mathcal{X}^n \to \mathcal{Y}$ that runs $Q$ a fixed number of times $k$ and returns the best output from these runs. Then $A$ is not $\hat{\varepsilon}$-DP for any $\hat{\varepsilon} < k\varepsilon$. Furthermore, for all $\lambda > 1$, $A$ is not $(\lambda, \hat{\varepsilon}(\lambda))$-Rényi DP for any $\hat{\varepsilon}(\lambda) < \varepsilon'(\lambda)$, where*

$$
\varepsilon'(\lambda) = k\varepsilon - \frac{k \cdot \log(1 + e^{-\varepsilon})}{\lambda - 1}.
$$

*Proof.* The base algorithm is simply randomized response. We will let $\mathcal{Y} = \{1, 2\}$ with the total order preferring 1, then 2. We will define a pair of distributions $Q$ and $Q'$ on $\{1, 2\}$ and then the base algorithm is simply set so that these are its output distributions on a pair of neighbouring inputs.

We let

$$
\begin{aligned}
Q &= \left(\frac{1}{1 + e^\varepsilon}, \frac{e^\varepsilon}{1 + e^\varepsilon}\right), \\
Q' &= \left(\frac{e^\varepsilon}{1 + e^\varepsilon}, \frac{1}{1 + e^\varepsilon}\right).
\end{aligned}
$$

Then $D_\infty(Q\|Q') = D_\infty(Q'\|Q) = \varepsilon$. Thus we can ensure that the base algorithm yielding this pair of distributions is $\varepsilon$-DP.

Now we look at the corresponding pair of distributions from repeating the base algorithm $k$ times. We have

$$A = \left(1 - \left(\frac{e^\varepsilon}{1+e^\varepsilon}\right)^k, \left(\frac{e^\varepsilon}{1+e^\varepsilon}\right)^k\right),$$

$$A' = \left(1 - \left(\frac{1}{1+e^\varepsilon}\right)^k, \left(\frac{1}{1+e^\varepsilon}\right)^k\right).$$

The first part of the result follows:

$$D_\infty(A\|A') \geq \log\left(\frac{\left(\frac{e^\varepsilon}{1+e^\varepsilon}\right)^k}{\left(\frac{1}{1+e^\varepsilon}\right)^k}\right) = k\varepsilon.$$

For all $\lambda > 1$,

$$e^{(\lambda-1)D_\lambda(A\|A')} \geq \left(\left(\frac{e^\varepsilon}{1+e^\varepsilon}\right)^k\right)^\lambda \cdot \left(\left(\frac{1}{1+e^\varepsilon}\right)^k\right)^{1-\lambda}$$

$$= e^{\varepsilon k\lambda} \cdot (1+e^\varepsilon)^{-k}$$

Hence

$$D_\lambda(A\|A') \geq \frac{\varepsilon k\lambda - k \cdot \log(1+e^\varepsilon)}{\lambda - 1} = k\varepsilon - \frac{k \cdot \log(1+e^{-\varepsilon})}{\lambda - 1}.$$

$\square$

The second part of Proposition 17 shows that this problem is not specific to pure DP. For $\lambda \geq 1 + 1/\varepsilon$, we have $\varepsilon'(\lambda) = \Omega(k\varepsilon)$, so we are paying linearly in $k$.

However, Proposition 17 is somewhat limited to pure $\varepsilon$-DP or at least $(\lambda, \varepsilon(\lambda))$-RDP with not-too-small values of $\lambda$. This is because the "bad" event is relatively low-probability. Specifically, the high privacy loss event has probability $(1 + e^{-\varepsilon})^{-k}$. This is small, unless $\varepsilon \geq \Omega(\log k)$.

We can change the example to make the bad event happen with constant probability. However, the base algorithm will also not be pure $\varepsilon$-DP any more. Specifically, we can replace the two distributions in Proposition 17 with the following:

$$Q = (1 - \exp(-1/k), \exp(-1/k)),$$
$$Q' = (1 - \exp(-\varepsilon_0 - 1/k), \exp(-\varepsilon_0 - 1/k)).$$

If we repeat this base algorithm a fixed number of times $k$, then the corresponding pair of distributions is given by

$$A = (1 - \exp(-1), \exp(-1)),$$
$$A' = (1 - \exp(-k\varepsilon_0 - 1), \exp(-k\varepsilon_0 - 1)).$$

Now we have $D_\infty(A\|A') = k\varepsilon_0$ and the bad event happens with probability $e^{-1} \approx 0.36$. On the other hand, $D_\infty(Q\|Q') = \varepsilon_0$ like before, but $D_\infty(Q'\|Q) = \log(1 - \exp(-\varepsilon_0 - 1/k)) - \log(1 - \exp(-1/k)) \approx \log(k\varepsilon_0 + 1)$. But we still have a good guarantee in terms of Rényi divergences. In particular, $D_1(Q'\|Q) \leq (\varepsilon_0 + 1/k)\log(k\varepsilon_0 + 1)$, and we can set $\varepsilon_0 \leq o(1/\log k)$ to ensure that we get reasonable $(\lambda, \varepsilon(\lambda))$-RDP guarantees for small $\lambda$.

At a higher level, it should not be a surprise that this negative example is relatively brittle. Our positive results show that it only takes a very minor adjustment to the number of repetitions to obtain significantly tighter privacy guarantees for hyperparameter tuning than what one would obtain from naive composition. In particular, running a fixed number of times $k$ versus running Poisson$(k)$ times is not that different, but our positive results show that it already circumvents this problem in general.

We also remark that composition behaves differently in the low-order RDP or approx DP regime relative to the pure-DP or high-order RDP regime covered by Proposition 17. Thus the naïve composition baseline we compare to is also shifting from basic composition ($\varepsilon$-DP becomes $k\varepsilon$-DP under $k$-fold repetition (Dwork et al., 2006b)) to advanced composition ($\varepsilon$-DP becomes ($O(\varepsilon \cdot \sqrt{k \cdot \log(1/\delta)}), \delta$)-DP (Dwork et al., 2010)). Proving tightness of basic composition is easy, but proving tightness for advanced composition is non-trivial (in general, it relies on the machinery of fingerprinting codes (Bun et al., 2014)). This means it is not straightforward to extend Proposition 17 to this regime.

### D.2 Tight example for conditional sampling.

We are also interested in the tightness of our generic results. We begin by studying the conditional sampling approach outlined in Appendix C. This approach is simpler and it is therefore easier to give a tight example. This also proves to be instructive for the random repetition approach in Section 3.

Our tight example for the conditional sampling approach consists of a pair of distributions $Q$ and $Q'$. These should be thought of as the output distributions of the algorithm $Q$ on two neighbouring inputs. The distributions are supported on only three points. Such a small output space seems contrived, but it should be thought of as representing a partition of a large output space into three sets. The first set is where the privacy loss is large and the second set is where the privacy loss is very negative, while the third set is everything in between.

Fix $s, t > 0$ and $a \in [0, 1/4]$. Note $(1 - 2a)^{-1} \le e^{3a}$. Let

$$
\begin{aligned}
Q &= \left(a \cdot e^{-s}, a \cdot e^{-s}, 1 - 2a \cdot e^{-s}\right), \\
Q' &= \left(a \cdot e^{-s-t}, a, 1 - a - a \cdot e^{-s-t}\right), \\
S &= \{1, 2\} \subset \{1, 2, 3\}, \\
Q_S &= \left(\frac{1}{2}, \frac{1}{2}\right), \\
Q'_S &= \left(\frac{e^{-s-t}}{1 + e^{-s-t}}, \frac{1}{1 + e^{-s-t}}\right).
\end{aligned}
$$

Intuitively, the set $S$ corresponds to the outputs that (1) have large privacy loss or (2) very negative privacy loss and we exclude (3) the outputs with middling privacy loss. Once we condition on $S$ we still have the outputs (1) with large privacy loss, but that privacy loss is further increased because of the renormalization. Specifically, the negative privacy loss means the renormalization constants are very different $- Q(S) = a \cdot 2e^{-s} \ll Q'(S) = a \cdot (1 + e^{-s-t}) -$ if $s$ is large. In effect, the negative privacy loss becomes a positive privacy loss that is added to the already large privacy loss.

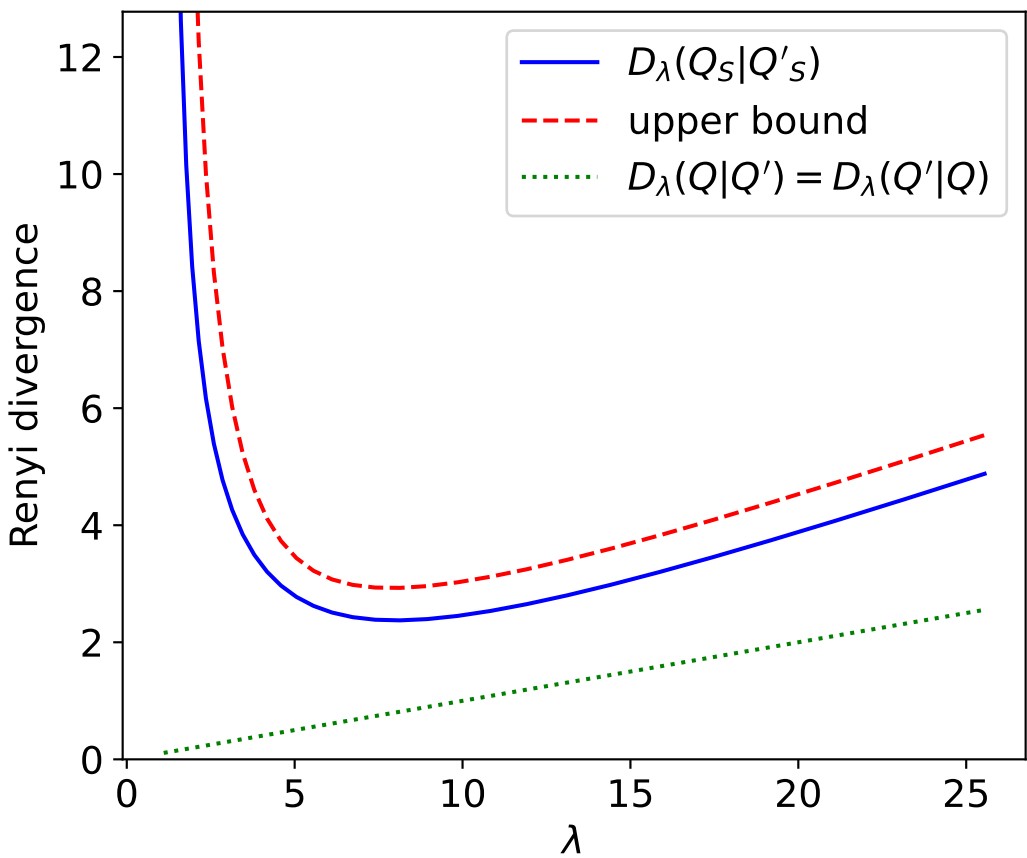

Figure 8: Upper and lower bounds for Rényi DP of conditional sampling. For each $\lambda$, we pick the parameters $s$ and $t$ such that $D_\lambda(Q\|Q') = D_\lambda(Q'\|Q) = 0.1 \cdot \lambda$ and we plot the upper bound from the second inequation in Corollary 16 along with the exact value of $D_\lambda(Q_S\|Q'_S)$.

We make the above intuition more precise by calculating the various quantities. For all $\lambda > 1$, we have

$$Q(S) = 2a \cdot e^{-s},$$

$$e^{(\lambda-1)D_\lambda(Q\|Q')} = a \cdot e^{-s\lambda-(s+t)(1-\lambda)} + a \cdot e^{-s\lambda} + (1-2a\cdot e^{-s})^\lambda (1-a-a\cdot e^{-s-t})^{1-\lambda}$$

$$\leq a \cdot e^{(\lambda-1)t-s} + ae^{-s\lambda} + e^{-2ae^{-s}\lambda}(1-2a)^{1-\lambda}$$

$$\leq a \cdot e^{(\lambda-1)t-s} + a + e^{3a(\lambda-1)}$$

$$\approx a \cdot e^{(\lambda-1)t-s} \qquad\qquad \text{(assuming } t \text{ is large)}$$

$$= \frac{1}{2}Q(S) \cdot e^{(\lambda-1)t},$$

$$\implies D_\lambda(Q\|Q') \lesssim t - \frac{\log(2/Q(S))}{\lambda-1},$$

$$e^{(\lambda-1)D_\lambda(Q'\|Q)} = a \cdot e^{-s(1-\lambda)-(s+t)\lambda} + a \cdot e^{-s(1-\lambda)} + (1-2a\cdot e^{-s})^{1-\lambda}(1-a-a\cdot e^{-s-t})^\lambda$$

$$\leq a \cdot e^{-t\lambda-s} + a \cdot e^{s(\lambda-1)} + e^{3a\cdot e^{-s}\cdot(\lambda-1)} \cdot e^{-a\lambda}$$

$$\approx a \cdot e^{s(\lambda-1)} \qquad\qquad \text{(assuming } s \text{ is large)}$$

$$= \frac{1}{2}Q(S) \cdot e^{s\lambda}$$

$$\implies D_\lambda(Q'\|Q) \lesssim s + \frac{s - \log(2/Q(S))}{\lambda-1},$$

$$e^{(\lambda-1)D_\lambda(Q_S\|Q'_S)} = 2^{-\lambda}(1 + e^{-s-t})^{\lambda-1}(e^{(\lambda-1)(s+t)} + 1)$$
$$\geq 2^{-\lambda} \cdot e^{(\lambda-1)(s+t)},$$
$$\implies D_\lambda(Q_S\|Q'_S) \geq s + t - \frac{\lambda \log 2}{\lambda - 1}.$$

We contrast this with our upper bound from the second part of Corollary 16:

$$D_\lambda(Q_S\|Q'_S) \leq D_\lambda(Q\|Q') + \frac{\lambda-2}{\lambda-1}D_{\lambda-1}(Q'\|Q) + \frac{2}{\lambda-1}\log\left(\frac{1}{Q(S)}\right).$$
$$\lesssim t - \frac{\log(2/Q(S))}{\lambda-1} + \frac{\lambda-2}{\lambda-1}\left(s + \frac{s - \log(2/Q(S))}{\lambda-2}\right) + \frac{2\log(1/Q(S))}{\lambda-1}$$
$$= s + t - \frac{2\log 2}{\lambda-1}.$$

This example shows that our upper bound is tight up to small factors, namely the lower order terms we ignore with $\approx$ and $\frac{\lambda-2}{\lambda-1}\log 2$. Figure 8 illustrates how the upper and lower bounds compare.

### D.3 TIGHTNESS OF OUR GENERIC RESULT.

Now we consider the setting from Section 3 where our base algorithm is run repeatedly a random number of times and the best result is given as output.

Let $Q : \mathcal{X}^n \to \mathcal{Y}$ denote the base algorithm. Assume $\mathcal{Y}$ is totally ordered. Let $K \in \mathbb{N}$ be a random variable and let $f(x) = \mathbb{E}\left[x^K\right]$ be its probability generating function. Define $A : \mathcal{X}^n \to \mathcal{Y}$ to be the algorithm that runs $Q$ repeatedly $K$ times and returns the best output.

For a tight example, we again restrict our attention to distributions supported on three points:

$$Q = Q(x) = (1 - b - c, b, c),$$
$$Q' = Q(x') = (1 - b' - c', b', c'),$$
$$A = A(x) = (1 - f(b + c), f(b + c) - f(c), f(c)),$$
$$A' = A(x') = (1 - f(b' + c'), f(b' + c') - f(c'), f(c')).$$

Here the total ordering prefers the first option (corresponding to the first coordinate probability), then the second, and then the third, which implies the expressions for $A$ and $A'$. Note that the probability values are not necessarily ordered the same way as the ordering on outcomes.

Now we must set these four values to show tightness of our results.

We make the first-order approximation

$$A \approx (1 - f(b + c), f'(c) \cdot b, f(c)),$$
$$A' \approx (1 - f(b' + c'), f'(c') \cdot b', f(c')).$$

We take this approximation and get

$$e^{(\lambda-1)D_\lambda(A\|A')} \gtrsim (f'(c) \cdot b)^\lambda \cdot (f'(c') \cdot b')^{1-\lambda}$$
$$= e^{(\lambda-1)D_\lambda(b\|b')} \cdot (f'(c))^\lambda \cdot (f'(c'))^{1-\lambda}$$
$$\approx e^{(\lambda-1)D_\lambda(Q\|Q')} \cdot (f'(c))^\lambda \cdot (f'(c'))^{1-\lambda},$$

where the final approximation assumes that the second term is dominant in the equation

$$e^{(\lambda-1)D_\lambda(Q\|Q')} = (1 - b - c)^\lambda \cdot (1 - b' - c')^{1-\lambda} + b^\lambda \cdot (b')^{1-\lambda} + (c)^\lambda \cdot (c')^{1-\lambda}.$$

Contrast this with our upper bound (Lemma 7), which says

$$e^{(\lambda-1)D_\lambda(A\|A')} \leq e^{(\lambda-1)D_\lambda(Q\|Q')} \cdot (f'(q))^\lambda \cdot (f'(q'))^{1-\lambda},$$

where $q$ and $q'$ are arbitrary postprocessings of $Q$ and $Q'$. In particular, we can set the values so that $q = c$ and $q' = c'$. This is not a formal proof, since we make imprecise approximations. But it illustrates that our main generic result (Lemma 7) is tight up to low order terms.

## D.4 Selection & Lower Bounds

Private hyperparameter tuning is a generalization of the private selection problem. In the private selection problem we are given a utility function $u : \mathcal{X}^n \times [m] \to \mathbb{R}$ which has sensitivity 1 in its first argument – i.e., for all neighbouring $x, x' \in \mathcal{X}^n$ and all $j \in [m] = \{1, 2, \cdots, m\}$, we have $|u(x, j) - u(x', j)| \leq 1$. The goal is to output and approximation to $\arg \max_{j \in [m]} u(x, j)$ subject to differential privacy.

The standard algorithm for private selection is the exponential mechanism (McSherry & Talwar, 2007). The exponential mechanism is defined by

$$\forall j \in [m] \quad \mathbb{P}\left[M(x) = j\right] = \frac{\exp\left(\frac{\varepsilon}{2} u(x, j)\right)}{\sum_{\ell \in [m]} \exp\left(\frac{\varepsilon}{2} u(x, \ell)\right)}.$$

It provides $(\varepsilon, 0)$-DP and, at the same time, $\frac{1}{8}\varepsilon^2$-zCDP (Rogers & Steinke, 2021). On the utility side, we have the guarantees

$$\mathbb{E}\left[u(x, M(x))\right] \geq \max_{j \in [m]} u(x, j) - \frac{2}{\varepsilon} \log m,$$

$$\mathbb{P}\left[u(x, M(x)) \geq \max_{j \in [m]} u(x, j) - \frac{2}{\varepsilon} \log\left(\frac{m}{\beta}\right)\right] \geq 1 - \beta$$

for all inputs $x$ and all $\beta > 0$ (Bassily et al., 2021, Lemma 7.1) (Dwork & Roth, 2014, Theorem 3.11).

It is also well-known that the exponential mechanism is optimal up to constants. That is, $(\varepsilon, 0)$-DP selection entails an additive error of $\Omega(\log(m)/\varepsilon)$ (Steinke & Ullman, 2017).

Our results can be applied to the selection problem. The base algorithm $Q(x)$ will simply pick an index $j \in [m]$ uniformly at random and the privately estimate $u(x, j)$ by adding Laplace or Gaussian noise $\xi$ and output the pair $(j, u(x, j) + \xi)$. The total order on the output space $[m] \times \mathbb{R}$ simply selects for the highest estimated utility (breaking ties arbitrarily). If we take $\xi$ to be Laplace noise with scale $1/\varepsilon$, then $Q$ is $(\varepsilon, 0)$-DP.

Applying Corollary 3 yields a $((2 + \eta)\varepsilon, 0)$-DP algorithm $A$ with the following utility guarantee. Let $K$ be the number of repetitions and let $(j_1, u(x, j_1) + \xi_1), \cdots, (j_K, u(x, j_K) + \xi_K)$ be the outputs from the runs of the base algorithm. The probability that the repeated algorithm $A$ will consider $j_* := \arg \max_{j \in [m]} u(x, j)$ is $\mathbb{P}\left[j_* \in \{j_1, \cdots, j_K\}\right] = 1 - \underset{K}{\mathbb{E}}\left[\prod_{k \in [K]} \mathbb{P}\left[j_* \neq j_k\right]\right] = 1 - f(1 - 1/m)$, where $f(x) = \mathbb{E}\left[x^K\right]$ is the probability generating function of the number of repetitions $K$. For each noise sample, we have $\forall k \ \mathbb{P}\left[|\xi_k| \leq t\right] \geq 1 - e^{-\varepsilon t}$ for all $t > 0$. Thus, for all $t > 0$, the probability that all noise samples are smaller than $t$ is $\mathbb{P}\left[\forall k \in [K] \ |\xi_k| \leq t\right] = f(1 - e^{-\varepsilon t})$. By a union bound, we have $\mathbb{P}\left[u(x, M(x)) \geq u(x, j_*) - 2t\right] \geq f(1 - e^{-\varepsilon t}) - f(1 - 1/m)$ for all $t > 0$. Setting $\eta = 0$, yields $f(x) = \frac{\log(1 - (1 - \gamma)x)}{\log \gamma}$, so $\mathbb{P}\left[u(x, M(x)) \geq u(x, j_*) - 2t\right] \geq \frac{1}{\log(1/\gamma)} \log\left(\frac{1 + \frac{1 - \gamma}{\gamma} \cdot \frac{1}{m}}{1 + \frac{1 - \gamma}{\gamma} \cdot e^{-\varepsilon t}}\right)$. Now set $t = \log(m^{10} - 1)/\varepsilon$ and $\gamma = \frac{1}{\exp(\varepsilon t) + 1} = m^{-10}$ so that $\frac{1 - \gamma}{\gamma} e^{-\varepsilon t} = 1$ and $\frac{1 - \gamma}{\gamma} \frac{1}{m} = m^9 - \frac{1}{m}$. Then $\mathbb{P}\left[u(x, M(x)) \geq u(x, j_*) - \frac{20}{\varepsilon} \log m\right] \geq \frac{1}{10 \log m} \log\left(\frac{m^9 + 1 - 1/m}{2}\right) \geq \frac{9}{10} - \frac{1}{10 \log_2 m}$. That is, we can match the result of the exponential mechanism up to (large) constants. In particular, this means that the lower bounds for selection translate to our results – i.e., our results are tight up to constants.

## E Extending our Results to Approximate DP

Our results are all in the framework of Rényi DP. A natural question is what can be said if the base algorithm instead only satisfies approximate DP – i.e., $(\varepsilon, \delta)$-DP with $\delta > 0$. Liu & Talwar (2019) considered these questions and gave several results. We now briefly show how to our results can be extended in a black-box fashion to this setting.

We begin by defining approximate Rényi divergences and approximate Rényi DP:

**Definition 18** (Approximate Rényi Divergence). *Let $P$ and $Q$ be probability distributions over $\Omega$. Let $\lambda \in [1, \infty]$ and $\delta \in [0, 1]$. We define*

$$\mathrm{D}_\lambda^\delta \left( P \| Q \right) = \inf \left\{ \mathrm{D}_\lambda \left( P' \| Q' \right) : P = (1 - \delta)P' + \delta P'', Q = (1 - \delta)Q' + \delta Q'' \right\},$$

*where $P = (1 - \delta)P' + \delta P''$ denotes the fact that $P$ can be expressed as a convex combination of two distributions $P'$ and $P''$ with weights $1 - \delta$ and $\delta$ respectively.*

**Definition 19** (Approximate Rényi Differential Privacy). *A randomized algorithm $M : \mathcal{X}^n \to \mathcal{Y}$ is $\delta$-approximately $(\lambda, \varepsilon)$-Rényi differentially private if, for all neighbouring pairs of inputs $x, x' \in \mathcal{X}^n$, $\mathrm{D}_\lambda^\delta \left( M(x) \| M(x') \right) \leq \varepsilon$.*

Definition 19 is an extension of the definition of approximate zCDP (Bun & Steinke, 2016). Some remarks about the basic properties of approximate RDP are in order:

- $(\varepsilon, \delta)$-DP is equivalent to $\delta$-approximate $(\infty, \varepsilon)$-RDP.

- $(\varepsilon, \delta)$-DP implies $\delta$-approximate $(\lambda, \frac{1}{2}\varepsilon^2\lambda)$-RDP for all $\lambda \in (1, \infty)$.

- $\delta$-approximate $(\lambda, \varepsilon)$-RDP implies $(\hat{\varepsilon}, \hat{\delta})$-DP for

$$\hat{\delta} = \delta + \frac{\exp((\lambda - 1)(\hat{\varepsilon} - \varepsilon))}{\lambda} \cdot \left( 1 - \frac{1}{\lambda} \right)^{\lambda - 1}.$$

- $\delta$-approximate $(\lambda, \varepsilon)$-Rényi differential privacy is closed under postprocessing.

- If $M_1$ is $\delta_1$-approximately $(\lambda, \varepsilon_1)$-Rényi differentially private and $M_2$ is $\delta_2$-approximately $(\lambda, \varepsilon_2)$-Rényi differentially private, then their composition is $(\delta_1 + \delta_2)$-approximately $(\lambda, \varepsilon_1 + \varepsilon_2)$-RDP.

Our results for Rényi DP can be extended to approximate Rényi DP by the following Lemma.

**Lemma 20.** *Assume $\mathcal{Y}$ is a totally ordered set. For a distribution $Q$ on $\mathcal{Y}$ and a random variable $K$ supported on $\mathbb{N} \cup \{0\}$, define $A_Q^K$ as follows. First we sample $K$. Then we sample from $Q$ independently $K$ times and output the best of these samples. This output is a sample from $A$.*

*Let $Q, Q', Q_{1-\delta_0}, Q_{\delta_0}, Q'_{1-\delta_0}, Q'_{\delta_0}$ be distributions on $\mathcal{Y}$ satisfying $Q = (1 - \delta_0)Q_{1-\delta_0} + \delta_0 Q_{\delta_0}$ and $Q' = (1 - \delta_0)Q'_{1-\delta_0} + \delta_0 Q'_{\delta_0}$. Let $K$ be a random variable on $\mathbb{N} \cup \{0\}$ and let $f(x) = \mathbb{E}\left[x^K\right]$ be the probability generating function of $K$. Define a random variable $K'$ on $\mathbb{N} \cup \{0\}$ by $\mathbb{P}\left[K' = k\right] = \mathbb{P}\left[K = k\right] \cdot (1 - \delta_0)^k / f(1 - \delta_0).$[12]*

*Then, for all $\lambda \geq 1$, we have*

$$\mathrm{D}_\lambda^\delta \left( A_Q^K \| A_{Q'}^K \right) \leq \mathrm{D}_\lambda \left( A_{Q_{1-\delta_0}}^{K'} \Big\| A_{Q'_{1-\delta_0}}^{K'} \right)$$

*where*

$$\delta = 1 - f(1 - \delta_0).$$

How do we use this lemma? We should think of $A_Q^K$ as representing the algorithm we want to analyze. The base algorithm $Q$ satisfies $\delta_0$-approximate $(\lambda, \varepsilon)$-RDP. The above lemma says it suffices to analyze the algorithm $A_{\tilde{Q}}^{K'}$ where $\tilde{Q}$ satisfies $(\lambda, \varepsilon)$-RDP. We end up with a $\delta$-approximate RDP result, where the final $\delta$ depends on $\delta_0$ and the PGF of $K$.

As an example, we can combine Lemma 20 with Theorem 6 to obtain the following result for the approximate case.

**Corollary 21.** *Let $Q : \mathcal{X}^n \to \mathcal{Y}$ be a randomized algorithm satisfying $(\varepsilon_0, \delta_0)$-DP. Assume $\mathcal{Y}$ is totally ordered. Let $\mu > 0$.*

*Define an algorithm $A : \mathcal{X}^n \to \mathcal{Y}$ as follows. Draw $K$ from a Poisson distribution with mean $\mu$. Run $Q(x)$ repeatedly $K$ times. Then $A(x)$ returns the best value from the $K$ runs. (If $K = 0$, $A(x)$ returns some arbitrary output independent from the input $x$.)*

---

[12]Note that the PGF of $K'$ is given by $\mathbb{E}\left[x^{K'}\right] = \sum_{k=0}^{\infty} x^k \mathbb{P}\left[K = k\right](1 - \delta_0)^k / f(1 - \delta_0) = f(x(1 - \delta_0))/f(1 - \delta_0).$

*For all $\lambda \leq 1 + \frac{1}{e^\varepsilon - 1}$, the algorithm A satisfies $\delta'$-approximate $(\lambda, \varepsilon')$-RDP where*

$$\varepsilon' = \varepsilon_0 + (e^{\varepsilon_0} - 1) \cdot \log \mu,$$
$$\delta' = 1 - e^{-\mu \cdot \delta_0} \leq \mu \cdot \delta_0.$$

*Proof of Lemma 20.* For a distribution $P$ on $\mathcal{Y}$ and an integer $k \geq 0$, let $\max P^k$ denote the distribution on $\mathcal{Y}$ obtained by taking $k$ independent samples from $P$ and returning the maximum value per the total ordering on $\mathcal{Y}$. (If $k = 0$, this is some arbitrary fixed distribution.)

Using this notation, we can express $A_Q^K$ as a convex combination:

$$A_Q^K = \sum_{k=0}^{\infty} \mathbb{P}[K = k] \max Q^k.$$

Suppose $P = (1 - \delta)P' + \delta P''$ is a convex combination. We can view sampling from $P$ as a two-step process: first we sample a Bernoulli random variable $B \in \{0, 1\}$ with expectation $\delta$; if $B = 0$, we return a sample from $P'$ and, if $B = 1$, we return a sample from $P''$. Thus, if we draw $k$ independent samples from $P$ like this, then with probability $(1 - \delta)^k$ all of these Bernoullis are 0 and we generate $k$ samples from $P'$; otherwise, we generate some mix of samples from $P'$ and $P''$. Hence we can write $\max P^k = (1 - \delta)^k \max(P')^k + (1 - (1 - \delta)^k)P'''$ for some distribution $P'''$.

It follows that we can express

$$\begin{aligned}
A_Q^K &= \sum_{k=0}^{\infty} \mathbb{P}[K = k] \max Q^k \\
&= \sum_{k=0}^{\infty} \mathbb{P}[K = k] \left( (1 - \delta_0)^k \max Q_{1-\delta_0}^k + (1 - (1 - \delta_0)^k)P_k \right) \\
&= f(1 - \delta_0)A_Q^{K'} + (1 - f(1 - \delta_0))P_*
\end{aligned}$$

for some distributions $P_0, P_1, \cdots$ and $(1 - f(1 - \delta_0))P_* = \sum_{k=0}^{\infty} \mathbb{P}[K = k](1 - (1 - \delta_0)^k)P_k$.

Similarly, we can express $A_{Q'}^K = f(1 - \delta_0)A_{Q'_{1-\delta_0}}^{K'} + (1 - f(1 - \delta_0))P_*'$ for some distribution $P_*'$.

Using these convex combinations we have, by the definition of approximate Rényi divergence,

$$\mathrm{D}_\lambda^\delta \left( A_Q^K \big\| A_{Q'}^K \right) \leq \mathrm{D}_\lambda \left( A_{Q_{1-\delta_0}}^{K'} \big\| A_{Q'_{1-\delta_0}}^{K'} \right)$$

as $\delta = 1 - f(1 - \delta_0)$. $\qquad \square$

*Proof of Corollary 21.* Fix neighbouring inputs $x, x'$ and let $Q = Q(x)$ and $Q' = Q(x')$ be the corresponding pair of output distributions from the base algorithm. Then, in the notation of Lemma 20, $A(x) = A_Q^K$ and $A(x') = A_{Q'}^K$ for $K \sim \mathsf{Poisson}(\mu)$. Setting $\delta' = 1 - f(1 - \delta_0) = 1 - e^{-\mu \cdot \delta_0} \leq \mu \cdot \delta_0$, we have

$$\mathrm{D}_\lambda^{\delta'}(A(x) \| A(x')) = \mathrm{D}_\lambda^{\delta'}\left( A_Q^K \big\| A_{Q'}^K \right) \leq \mathrm{D}_\lambda \left( A_{Q_{1-\delta_0}}^{K'} \big\| A_{Q'_{1-\delta_0}}^{K'} \right)$$

where $K' \sim \mathsf{Poisson}(\mu \cdot (1 - \delta_0))$.

Now we apply Theorem 6 to the algorithm corresponding to the pair of distributions $A_{Q_{1-\delta_0}}^{K'}$ and $A_{Q'_{1-\delta_0}}^{K'}$. The base algorithm, corresponding to the pair $Q_{1-\delta_0}$ and $Q'_{1-\delta_0}$ satisfies $(\varepsilon_0, 0)$-DP. This yields

$$\mathrm{D}_\lambda \left( A_{Q_{1-\delta_0}}^{K'} \big\| A_{Q'_{1-\delta_0}}^{K'} \right) \leq \varepsilon_0 + \frac{\log((1 - \delta_0) \cdot \mu)}{\lambda - 1}$$

if $e^{\varepsilon_0} \leq 1 + 1/(\lambda - 1)$. Setting $\lambda = 1 + 1/(e^{\varepsilon_0} - 1)$ and applying monotonicity (Remark 5 and Lemma 10) and we obtain the result. $\qquad \square$

### E.1   TRUNCATING THE NUMBER OF REPETITIONS

Another natural question is what happens if we truncate the distribution of the number of repetitions $K$. For example, we may have an upper limit on the acceptable runtime. This does *not* require relaxing to approximate DP, as done by Liu & Talwar (2019).

Let $K$ be the non-truncated number of repetitions and let $f(x) = \mathbb{E}\left[x^K\right]$ be the PGF. Let $m \in \mathbb{N}$. Let $\tilde{K}$ be the truncated number of repetitions. That is, $\mathbb{P}\left[\tilde{K} = k\right] = \mathbb{I}[K \leq m] \cdot \mathbb{P}\left[K = k\right]/\mathbb{P}\left[K \leq m\right]$. Let $\tilde{f}(x) = \mathbb{E}\left[x^{\tilde{K}}\right]$ be the corresponding PGF.

We have $f'(x) = \sum_{k=1}^{\infty} k \cdot x^{k-1} \cdot \mathbb{P}\left[K = k\right]$ and $\tilde{f}'(x) = \sum_{k=1}^{m} k \cdot x^{k-1} \cdot \frac{\mathbb{P}[K=k]}{\mathbb{P}[K\leq m]}$. Hence, for $x \in [0,1]$,

$$0 \leq f'(x) - \tilde{f}'(x) \cdot \mathbb{P}\left[K \leq m\right]$$

$$= \sum_{k=m+1}^{\infty} k \cdot x^{k-1} \cdot \mathbb{P}\left[K = k\right]$$

$$\leq x^m \cdot \sum_{k=m+1}^{\infty} k \cdot \mathbb{P}\left[K = k\right]$$

$$= x^m \cdot \mathbb{E}\left[K \cdot \mathbb{I}[K > m]\right].$$

Now $\tilde{f}'(x) \cdot \mathbb{P}\left[K \leq m\right] = \sum_{k=1}^{m} k \cdot x^{k-1} \cdot \mathbb{P}\left[K = k\right] \geq x^{m-1} \cdot \mathbb{E}\left[K \cdot \mathbb{I}[K \leq m]\right]$. Thus

$$1 \leq \frac{f'(x)}{\tilde{f}'(x) \cdot \mathbb{P}\left[K \leq m\right]} \leq 1 + \frac{x^m \cdot \mathbb{E}\left[K \cdot \mathbb{I}[K > m]\right]}{x^{m-1} \cdot \mathbb{E}\left[K \cdot \mathbb{I}[K \leq m]\right]} = 1 + x \cdot \frac{\mathbb{E}\left[K \cdot \mathbb{I}[K > m]\right]}{\mathbb{E}\left[K\right] - \mathbb{E}\left[K \cdot \mathbb{I}[K > m]\right]}$$

Now we can bound the quantity of interest in Lemma 7: For all $q, q' \in [0,1]$, we have

$$\tilde{f}'(q)^\lambda \cdot \tilde{f}'(q')^{1-\lambda} \leq \left(\frac{f'(q)}{\mathbb{P}\left[K \leq m\right]}\right)^\lambda \cdot \left(\frac{f'(q')}{\mathbb{P}\left[K \leq m\right] \cdot \left(1 + \frac{\mathbb{E}[K \cdot \mathbb{I}[K > m]]}{\mathbb{E}[K] - \mathbb{E}[K \cdot \mathbb{I}[K > m]]}\right)}\right)^{1-\lambda}$$

$$= f'(q)^\lambda \cdot f'(q')^{1-\lambda} \cdot \frac{1}{\mathbb{P}\left[K \leq m\right]} \cdot \left(1 + \frac{\mathbb{E}\left[K \cdot \mathbb{I}[K > m]\right]}{\mathbb{E}\left[K\right] - \mathbb{E}\left[K \cdot \mathbb{I}[K > m]\right]}\right)^{\lambda-1}$$

This gives us a generic result for truncated distributions:

**Lemma 22** (Generic Bound (cf. Lemma 7) for Truncated distributions)**.** *Fix $\lambda > 1$ and $m \in \mathbb{N}$. Let $K$ be a random variable supported on $\mathbb{N} \cup \{0\}$. Let $f : [0,1] \to \mathbb{R}$ be the probability generating function of $K$ – i.e., $f(x) := \sum_{k=0}^{\infty} \mathbb{P}\left[K = k\right] \cdot x^k$.*

*Let $Q$ and $Q'$ be distributions on $\mathcal{Y}$. Assume $\mathcal{Y}$ is totally ordered. Define a distribution $A$ on $\mathcal{Y}$ as follows. First we sample $\tilde{K}$ which is $K$ conditioned on $K \leq m$ – i.e. $\mathbb{P}\left[\tilde{K} = k\right] = \mathbb{P}\left[K = k | K \leq m\right]$. Then we sample from $Q$ independently $\tilde{K}$ times and output the best of these samples.[13] This output is a sample from $A$. We define $A'$ analogously with $Q'$ in place of $Q$.*

*Then*

$$\mathrm{D}_\lambda\left(A \| A'\right) \leq \mathrm{D}_\lambda\left(Q \| Q'\right) + \frac{1}{\lambda - 1} \log\left(f'(q)^\lambda \cdot f'(q')^{1-\lambda}\right) + \frac{\log\left(\frac{1}{1 - \mathbb{P}[K > m]}\right)}{\lambda - 1} + \log\left(1 + \frac{\mathbb{E}\left[K \cdot \mathbb{I}[K > m]\right]}{\mathbb{E}\left[K\right] - \mathbb{E}\left[K \cdot \mathbb{I}[K > m]\right]}\right),$$
(9)

*where $q$ and $q'$ are probabilities attained by applying the same arbitrary postprocessing to $Q$ and $Q'$ respectively – i.e., there exists a function $g : \mathcal{Y} \to [0,1]$ such that $q = \mathbb{E}_{X \leftarrow Q}\left[g(X)\right]$ and $q' = \mathbb{E}_{X' \leftarrow Q'}\left[g(X')\right]$.*

This will give almost identical bounds to using the non-truncated distribution as long as $\mathbb{P}\left[K > m\right] \ll 1$ and $\mathbb{E}\left[K \cdot \mathbb{I}[K > m]\right] \ll \mathbb{E}\left[K\right]$, which should hold for large enough $m$.

---

[13] If $\tilde{K} = 0$, the output can be arbitrary, as long as it is the same for both $A$ and $A'$.

