# OpenReview forum: "Hyperparameter Tuning with Renyi Differential Privacy"
_ICLR.cc/2022/Conference — ICLR 2022 Oral_

### Official Review · Reviewer_SYbz · 2021-11-02

**Correctness:** 4
**Technical Novelty And Significance:** 4
**Empirical Novelty And Significance:** 4
**Recommendation:** 10
**Confidence:** 5

**Main Review:**

DP-focused machine learning systems would benefit from this work, as most of them require hyperparameter tuning. The algorithms provided in this work help us avoid a large cost of composition and perform tuning at much smaller privacy loss. In addition, the generic bound allows us to be flexible on the distribution of the number of runs ($K$), though it requires some effort to translate the logarithmic term into something usable. I have checked all the proofs and there are no critical errors.

The authors have sufficiently compared their method with previous approaches. Specifically, using the main results, the authors show that their method extends and improves upon the work of Liu and Talwar (2019). The authors also compare their method with the exponential mechanism-based method (Gupta et al., 2010; Theorem 10.2) and show in Section D.4 that both methods have the same lower bounds of the utility guarantees up to constants.

Nonetheless, the authors might want to compare their method with the noise perturbation method proposed by Chaudhuri et al. (2013). This method requires the “stability” condition on the training procedure, which might not be tractable for neural networks. But for linear classifiers, the algorithm (Algorithm 1) only adds noises of size $O(1/n\varepsilon)$ to the scores, which is quite attractive for training on large datasets.

The paper is mostly written with DP-SGD in mind, but it does not consider when the model's evaluation on a hold-out validation set is incorporated in the base algorithm $Q$, which is a common practice in training an ML model. From an application point of view, the authors might want to discuss a bit how the model’s evaluation (in classification or regression) can be made DP or RDP as a part of $Q$.

I see that distributions with finite support (e.g. the truncated binomial) are not considered in this study. But in practice, one might want to limit the number of hyperparameter searches (e.g. due to computational constraints), so such distributions might come into play. Have the authors performed some experiments on these against the truncated negative binomial and Poisson distribution? I wonder if the privacy-utility tradeoff is better when restricting to finite support.

Could the authors comment on how to determine an appropriate size of the candidate set ($m$)? A simple heuristic is $m=\mathbb{E}[K]$ but the authors might have something better in mind.

### **Specific comments**
* Page 2: In Eq (2), the special case where $\lambda=1$ should be mentioned here, as it is also in the range of $\hat\lambda$ in the main results.
* Page 6: When I tried to derive (6) from (5), I got an extra $-\rho\eta$ term from rewriting
$$\epsilon+\rho(\lambda-1)+(1+\eta)\left(1-\frac1{\hat\lambda}\right)\hat\epsilon=\rho\lambda+\rho(1+\eta)(\hat\lambda-1)=\rho(\lambda-1)+\rho\hat\lambda(1+\eta)-\rho\eta,$$
combining this with the rest of the terms, and then choosing $\lambda$ and $\lambda'$ so that the equality holds in the following inequality:
$$ \left(\rho(\lambda-1) + \frac{\log\mathbb{E}[K]}{\lambda-1} \right)+\left(\rho\hat\lambda(1+\eta) + \frac{(1+\eta)\log(1/\gamma)}{\hat\lambda}\right)- \rho\eta \\
  \geq  2\sqrt{\rho\log\mathbb{E}[K]} +2(1+\eta)\sqrt{\rho\log(1/\gamma)}-\rho\eta. $$
To clarify this, I think the authors should provide the proof of Corollary 4 somewhere in the Appendix, as I find the bound to be non-trivial.
* Page 7: In Lemma 7, “$q$ and $q'$ are arbitrary probabilities” is misleading. When I saw the statement for the first time, I read this as “$q$ and $q'$ can be anything in $[0,1]$”, but the proof indicates that they take specific forms in order for the Lemma to hold true. One way to resolve this issue is by directly stating the definition of $q$ and $q'$ after (7).
* Line 1-2 in Page 8:
> Vaguely, $f'$ being smooth corresponds to the distribution $K$ being spread out (i.e. far from being a point mass).
>
This is not true; if $X$ is a point mass at $1$ i.e. $\Pr{[X=1]}=1$, then the PGF of $X$ is $f(x)=x$, and so $f'(x)=1$ which is smoother than any polynomial.
Looking at the definition of PGF, the smoothness of $f'$ should depend on the right-tail heaviness of the distribution of $K$. Specifically, a heavier right tail corresponds to a faster growth rate of $f'$, which in turn leads to a larger RHS of Eq. (7) when $q > q'$. This observation is in line with the privacy-utility tradeoff: more probabilities of sampling a large $K$ (i.e. more utility) lead to a larger privacy loss.
* Page 18: when $p=r=\infty$, then the value of $r/p$ is unclear. What is the convention for this case?
* Page 23: It is mentioned below the definition of $Q,Q',A,A'$ that “the total ordering prefers the first option (corresponding to the first coordinate probability)” which should refer to the ones with the probabilities $1-b-c$ and $1-b'-c'$. In other words, we assume that $1-b-c>b$ and $1-b'-c'>b'$. However, the approximations below suggest that $b>c>1-b-c$ and $b'>c'>1-b'-c'$.
* Page 25: The proof of Lemma 20 is missing. Does it appear in Bun & Steinke (2016)?

### **Minor corrections**
* Page 6: In Remark 5: “$(\lambda_2,\varepsilon)$-RDP implies $(\lambda_2,\varepsilon)$-RDP” $\rightarrow$ “$(\lambda_2,\varepsilon)$-RDP implies $(\lambda_1,\varepsilon)$-RDP” and “ann” $\rightarrow$ “any”.
* Page 14: In the footnote, “experssion” $\rightarrow$ “expression”
* Page 16: In the definition of $g(y)$, $t$ should be $t^*$.
* Page 21: In the third display equation, the equal sign should be replaced with $>$.
* Page 23: First line in Section D.4: “hyperparamter” $\rightarrow$ “hyperparameter”

### **References**
Bun, M., & Steinke, T. (2016). Concentrated differential privacy: Simplifications, extensions, and lower bounds. TCC'16.
Chaudhuri, K., & Vinterbo, S.A. (2013). A Stability-based Validation Procedure for Differentially Private Machine Learning. NIPS'13.
Gupta, A., Ligett, K., McSherry, F., Roth, A., & Talwar, K. (2010). Differentially private combinatorial optimization. SODA '10.
Liu, J., & Talwar, K. (2019). Private selection from private candidates. STOC'19.


**Summary Of The Paper:**

The authors make the following contributions regarding differentially private hyperparameter tuning :
* As an example, the authors train an SVM with a weight penalty and show that, in presence of an outlier, a membership inference attack can be employed to infer from the hyperparameter whether or not the outlier was a part of the training set.
*The authors provide an algorithm for private hyperparameter tuning, which consists of running a learning algorithm that satisfies Rényi differential privacy (RDP) for a *random* number of times ($K$), each with a hyperparameter drawn uniformly at random from a finite candidate set. The authors first prove RDP guarantees of the algorithm when $K$ is sampled from a truncated negative binomial distribution and Poisson distribution. Then they proceed to prove RDP guarantees for any distribution of $K$ supported on $\mathbb{N}\cup \{0\}$ ¹.
The authors propose a way to measure the utility of the algorithms by looking at the expected quantile of the output. The results of their utility analysis, coupled with an experiment on MNIST, show that the algorithm with Poisson distribution performs better than those proposed by Liu and Talwar (2019) in an intermediate range of privacy budget ($\varepsilon$).

¹ From what I understand, we can obtain a tighter generic bound by going through the proof of Lemma 7. But the authors opt to use (7) since the postprocessing often leads to a bound that is independent of $q$ and $q'$ when plugging in a specific distribution.


**Summary Of The Review:**

This work provides careful privacy and utility analysis of private algorithms for hyperparameter tuning. All of the analyses and the proofs are sound, and the experiments give good comparisons between various distributions.  To make the methods widely applicable, the authors might want to comment on how the model’s evaluation on a hold-out validation set can be integrated into the DP workflow. Overall, this is a strong paper and I recommend it for publication.

---

> ### Author Response · Authors · 2021-11-17
> **Response to reviewer SYbz (part 1 / 2)**
>
> We thank the reviewer for their time and thorough comments. In the following, we respond inline to the reviewer’s comments.
>
> >_**Nonetheless, the authors might want to compare their method with the noise perturbation method proposed by Chaudhuri et al. (2013).**_
>
> Thank you for pointing out that we neglected to mention the work of Chaudhuri et al. Although these results are not directly comparable to ours, they are related. Our results are algorithm-agnostic in the sense that we only assume that the base algorithm is differentially private. Chaudhuri et al. follow an approach that assumes a stability property of the algorithm, which allows the application of a variant of the exponential mechanism to hyperparameter selection. Whereas the approach by Chaudhuri requires that one analyze the learning algorithm itself, our algorithm-agnostic approach is more readily applicable to modern machine learning systems, as they generally do not satisfy the stability criterion required. We have added a citation and brief comparison in our revised manuscript.
>
> >_**The paper is mostly written with DP-SGD in mind, but it does not consider when the model's evaluation on a hold-out validation set is incorporated in the base algorithm**_
>
> This is a good point, which we glossed over. The DP-SGD training procedure should only look at the training data and the evaluation should be conducted on held out data. From a privacy perspective, we can dedicate the same privacy budget ($\varepsilon$) to the evaluation as for the training, since these are disjoint sets. That is, we would evaluate the accuracy on the held out data at the end of each training run corresponding to a hyperparameter candidate value by, for instance, adding Laplace/Gaussian noise to preserve privacy. In this case, the privacy loss from evaluation would not be added to the loss from training since these are two separate datasets.
>
> Alternatively, the base algorithm could simply evaluate on the training data (and appeal to the generalization properties of differential privacy) and the held out data could be preserved for evaluating the final output from the repeated algorithm. We would still need to add noise to the evaluation, but now we must add this privacy loss to that of the training as it is not fresh data. We have added a footnote regarding this choice in Section 3.1.
>
> >_**I see that distributions with finite support (e.g. the truncated binomial) are not considered in this study. But in practice, one might want to limit the number of hyperparameter searches (e.g. due to computational constraints)**_
>
> We have added a result to Appendix E which covers truncating the distribution, although we did not consider this in our experiments. The message of this result is simply that the privacy guarantee would degrade gracefully if we did truncate.
> Computational efficiency is indeed an important consideration and this would speak to using the Poisson distribution, as it is highly concentrated, compared to the negative binomial distribution.
>
> >_**Could the authors comment on how to determine an appropriate size of the candidate set ($m$)? A simple heuristic is $m=\mathbb{E}[K]$ but the authors might have something better in mind.**_
>
> This is a good question. The choice should not be particularly brittle – i.e., choosing a slightly too large or too small search space should not dramatically alter the usefulness of the results. Our intuition is that the search space $m$ would be dictated by the application and then we would pick $K$ so that, say, $P[K \ge m] \ge 2/3$.

---

> > ### Author Response · Authors · 2021-11-17
> > **Response to reviewer SYbz (part 2 / 2)**
> >
> > Specific Comments.
> >
> > * The $\lambda=1$ case, which is the KL divergence, is defined in the appendix. Setting $\hat\lambda=1$ in our result is valid, but in this case the value of the divergence is multiplied by $0$, so it’s not actually necessary to quantify it. We have added a footnote.
> > * We have added a proof of Corollary 4 in the appendix and double-checked this calculation. The reviewer is correct that the result can be improved with an extra $-\eta\rho$ term.
> > * In Lemma 7 we have clarified that q and q’ are attained by applying the same arbitrary postprocessing to Q and Q’ respectively.
> > * The “smoothness” of the derivative of the PGF does depend on the right-tail heaviness of the distribution, but there are other factors involved. E.g. the negative binomial distribution has a very heavy right tail, but its PGF behaves quite well in this regard. There is an interplay between the PGF and the privacy guarantee. We have slightly elaborated this intuition, but ultimately the connection to the PGF is not particularly intuitive and we must rely on the formal analysis.
> > * When $p=r=\infty$ then the application of Holder’s inequality in the proof of Proposition 15 is simple – i.e. $\int_S Q(x)^\lambda Q’(x)^{1-\lambda} \mathrm{d}x \le (\max_x Q(x)/Q’(x))^\lambda \cdot \int_S Q’(x) \mathrm{d}x$. The expression $r\lambda-r/p = r(\lambda-1/p)$ should be evaluated as $\infty$ (the $r/p$ term is not significant in the limit). We have clarified this expression.
> > * The total ordering in the example on page 23 does not necessarily correspond to the ordering of probabilities. We have added a short remark on this.
> > * Page 25 / Appendix E: We apologize for this appendix not being complete. We have largely rewritten this with complete proofs and improved results.
> >
> > We would like to thank the reviewer for carefully reviewing our manuscript. We have applied all the minor corrections requested in the rest of your review.

---

> > > ### Comment · Reviewer_SYbz · 2021-11-17
> > > **Response**
> > >
> > > I wouldn't say that an extra $-\eta \rho$ term is strictly an improvement, since $\eta$ can be negative.
> > >
> > > I would like to thank the authors for responding; All of my concerns have been carefully addressed.

---

### Official Review · Reviewer_4Ysq · 2021-11-02

**Correctness:** 3
**Technical Novelty And Significance:** 4
**Empirical Novelty And Significance:** 3
**Recommendation:** 8
**Confidence:** 3

**Main Review:**

The paper is very well written and the analysis seems solid (though I did not check every line). The problem is important and the paper improves the state-of-the-art so its merits are clear.

My only critique:

As emhpasised in Sec. 3.2, the ‘strawman approach’ of fixing m does not work for pure eps-DP. However, I find this a bit misleading since you are not observing (eps,0)-DP of hyperparameter tuning but (eps,delta) (or RDP). So the delta might actually play a crucial role here. Allowing bit of delta in the DP bound, the randomness in choosing the number of repetition might perhaps be not that crucial. Or vice versa, in (eps,0), you would not get great gains from that randomness.

I suspect that this fact that you have to have the randomness in the number of repetitions actually is a requirement of the RDP analysis that you carry out. As far as I see, in Proposition 17 you claim it is not, but I do not fully see why that would be the case. The result of Proposition 17, i.e., that the RDP of k repetitions is not RDP for less than eps’(lambda), does not really give a ‘counter-example’ in the same  way as that result for pure epsilon. For example, suppose the underlying mechanism is eps=0.5 - DP. Then, you choose lambda = 1.9483. Then, eps - (log(1+exp(-eps))/(lambda-1)) ~ 1e-4 in which case the bound does not really say anything even for quite large numbers of k. And you can of course make that bound arbitrarily small by choosing lambda appropriately. I.e., if you let delta > 0, I think there is room for tighter analysis also for fixed k.

Could you comment on this tradeoff between lambda and eps in the bound of Proposition 17?

Could you give more intuition on why the randomness in number of repetitions k would be crucial or some other example that illustrates this?

Other:
- There is something strange at the bottom of p. 22: equation going over the line.


**Summary Of The Paper:**

The paper provides an considerable improvement to the DP analysis of hyperparameter tuning of DP algorithms (such as DP-SGD). The analysis is carried out using Rényi differential privacy (RDP), and the DP bounds are RDP bounds that contain the RDP parameters of the underlying mechanisms (that give the private candidates). Most importantly, the paper considerably improves the state-of-the-art of Liu and Talwar (2019). Also, a nice counterexample using SVMs is constructed, that shows the importance of this problem.


**Summary Of The Review:**

All in all I think this is a very nice analysis and improves the state-of-the-art. Also, this is a very important problem. With small modifications I think this ought to be accepted. My only critique: I am just not entirely convinced that randomness in the number of repetitions is crucial for having tight bounds for DP hyperparameter tuning, I hope the authors can clarify my concerns.

---

> ### Author Response · Authors · 2021-11-21
> **Response to reviewer 4Ysq**
>
> We thank the reviewer for their time and comments.
>
> >_** Could you comment on this tradeoff between lambda and eps in the bound of Proposition 17? Could you give more intuition on why the randomness in k would be crucial and/or some other example that illustrates this? **_
>
> The reviewer has astutely observed that the negative result for our strawman approach (Proposition 17) is somewhat specific to pure $\varepsilon$-DP or at least $(\lambda,\varepsilon(\lambda))$-RDP with not-too-small values of $\lambda$. This is because the "bad" event is relatively low-probability. Specifically, the high privacy loss event has probability $(1+e^{-\varepsilon})^{-k}$. This is small, unless $\varepsilon \ge \Omega( \log k)$.
>
> We can change the example to make the bad event happen with constant probability. However, the base algorithm will also not be pure $\varepsilon$-DP any more.
> Specifically, we can replace the two distributions in Proposition 17 with the following:
> $$Q=(1-\exp(-1/k),\exp(-1/k))$$
> $$Q’=(1-\exp(-\varepsilon_0-1/k),\exp(-\varepsilon_0-1/k))$$
> If we repeat this base algorithm a fixed number of times $k$, then the corresponding pair of distributions is given by
> $$A=(1-\exp(-1),\exp(-1))$$
> $$A’=(1-\exp(-k\varepsilon_0-1),\exp(-k\varepsilon_0-1)$$
> Now we have $\mathrm{D}_\infty(A\|A’)=k\varepsilon_0$ and the bad event happens with probability $e^{-1} \approx 0.36$.
> On the other hand, $\mathrm{D}_\infty(Q\|Q’)=\varepsilon_0$ like before, but $\mathrm{D}_\infty(Q’\|Q) = \log(1-\exp(-\varepsilon_0-1/k)) - \log(1-\exp(-1/k)) \approx \log(k\varepsilon_0+1)$.
> But we still have a good guarantee in terms of Renyi divergences. In particular, $\mathrm{D}_1(Q’\|Q)\le (\varepsilon_0+1/k)\log(k\varepsilon_0+1)$, and we can set $\varepsilon_0 \le o(1/\log k)$ to ensure that we get reasonable $(\lambda,\varepsilon(\lambda))$-RDP guarantees for small $\lambda$.
>
> At a higher level, it should not be a surprise that this negative example is relatively brittle. Our positive results show that it only takes a very minor adjustment to the number of repetitions to obtain significantly tighter privacy guarantees for hyperparameter tuning than what one would obtain from naive composition. In particular, running a fixed number of times $k$ versus running $\mathsf{Poisson}(k)$ times is not that different, but our positive results show that it already circumvents this problem in general.
>
> We have added these comments to the appendix after Proposition 17.
>
> >_** There is something strange at the bottom of p. 22: equation going over the line. **_
>
> We thank you for pointing this out. We fixed this typographical error in the revised manuscript.

---

> > ### Comment · Reviewer_4Ysq · 2021-11-23
> > **Thanks for the reply**
> >
> > Thanks for clarifying this and for adding comments about this to the paper. This modified example seems more convincing yet slightly artificial as these modified $Q$ and $Q'$ also depend on $k$. Perhaps this is a situation where one could possibly see the difference of having tight $(\varepsilon,\delta)$-bounds and RDP bounds. But as you write in Section 3.2: 'This negative result also extends to Rényi DP..', so you are only claiming you have this for RDP which looks correct. I think this is a really nice paper, I have raised my score by 1.

---

### Official Review · Reviewer_mxg8 · 2021-11-03

**Correctness:** 4
**Technical Novelty And Significance:** 3
**Empirical Novelty And Significance:** 3
**Recommendation:** 6
**Confidence:** 4

**Main Review:**

This paper considers an interesting and important problem: how hyperparameter tuning on private dataset can leak information, where it provides an intuitive SVM example. The paper then considers how to reduce the leakage. The problem is formalized in the following way: we pick a total number of runs $K$ from some distribution. Then, for each run  $k = 1, 2,\ldots, K$ , we pick an index $j_k \in [m]$ uniformly at random and run $M_{jk}$ . Then, at the end, we return the best of the $K$ outcomes. The paper proposes theoretical guarantees when $K$ comes from truncated negative binomial distribution, or Poission distribution, which strictly generalizes the previous results. Furthermore, it also proposes a new method of computing privacy leakage when $K$ comes from a general distribution, which should be of independent interest. Finally, the paper conducts empirical experiments to show the improvement if the new method.

My only concern for this paper is the problem formalization. First, for the current problem formulation, it is possible that the same parameter will be tried multiple times, which is definitely a waste of privacy. Not sure whether people will do it in practice. Second, the paper assumes the following scheme satisfies DP: randomly choose $j \in [m]$, and run $M_j$. Note that this is not equivalent with assuming each $M_j$ is DP, where the latter is more realistic. For example, in the hyperparameter tuning of DP-SGD, it easily holds that each run (with different clipping norm) satisfies DP. However, it is not clear to me whether randomly selecting one clipping norm, and then running DP-SGD is differentially private. The authors need to justify the relationship between these two assumptions. Naively speaking, the second can not lead to the first assumption. Please correct me if I miss something.

**Summary Of The Paper:**

This paper has made the following contributions. Firstly, this paper illustrates how simply tuning hyperparameters based on non-private training runs can leak private information. Second, this paper provides privacy guarantees for hyperparameter search procedures within the framework of Renyi Differential Privacy. Their results improve and extend the work of Liu and Talwar (STOC 2019).

**Summary Of The Review:**

Generally speaking, I recommend acceptance of this paper.

---

> ### Author Response · Authors · 2021-11-17
> **Response to reviewer mxg8**
>
> We thank the reviewer for their time and feedback. In the following, we respond inline to the reviewer’s concerns and questions. If the reviewer has any further questions, we would be happy to elaborate in a further response.
>
> >_**First, for the current problem formulation, it is possible that the same parameter will be tried multiple times, which is definitely a waste of privacy. Not sure whether people will do it in practice.**_
>
> Our problem formulation follows that of Liu & Talwar. We follow this formulation both for simplicity and for consistency with the prior work. It is indeed potentially inefficient to repeat some runs. However, we remark that, since the algorithms are randomized, running them multiple times does not entail that we have the same output multiple times.
>
> >_**Second, the paper assumes the following scheme satisfies DP: randomly choose $j\in [m]$, and run $M_j$. Note that this is not equivalent with assuming each $M_j$ is DP, where the latter is more realistic. For example, in the hyperparameter tuning of DP-SGD, it easily holds that each run (with different clipping norm) satisfies DP. However, it is not clear to me whether randomly selecting one clipping norm, and then running DP-SGD is differentially private. The authors need to justify the relationship between these two assumptions. Naively speaking, the second can not lead to the first assumption. Please correct me if I miss something.**_
>
> Differential privacy is "convex" in the sense that if $M_1,M_2,\cdots,M_m$ are each individually differentially private, then randomizing over these – i.e., running $M_J$ for a random $J \in [m]$ is also differentially private with the same parameters. This convexity property is important for the problem setup and we added a note emphasizing this in Section 3.3.

---

> > ### Comment · Reviewer_mxg8 · 2021-11-23
> > **Response**
> >
> > Thanks the authors for the rebuttal, which answers my questions.

---

### Official Review · Reviewer_CBQT · 2021-11-03

**Correctness:** 4
**Technical Novelty And Significance:** 3
**Empirical Novelty And Significance:** 3
**Recommendation:** 8
**Confidence:** 3

**Main Review:**

The problem studied here is a really important one, since it is directly related to actual deployment of DP models in real life settings. The proposed method seems solid and the experimental results look promising. I do have a few clarification questions:

1. How would this method be extended to be used with adaptive search methods? Grid search can be too time consuming/inefficient, especially when used with DP-SGD (as DP-SGD needs per-example gradients that are expensive to obtain) and there are some reinforcement learning based methods that can yield optimal hyper parameters much faster than grid search. Can this method be modified, maybe with a higher privacy expenditure, to be used in those cases?

2. What kind of real-life attack scenario would you envision that could actually use the best models obtained by hyper parameter tuning to extract information. Basically, what type/how much information about a model do the set of optimal hyper parameters leak?



**Summary Of The Paper:**

This paper studies the problem of hyper parameter tuning in the setting of differentially private training of machine learning models. The paper first shows that the hyper parameters used to train a model and the corresponding utility can leak information about training (through experimenting with outliers). They then go on to introduce and analyze ways that would help release DP models trained with the "best" set of hyper parameters, with small privacy leakage. If for hyper parameter tuning, $m$ models are trained with DP, then the leakage for releasing  the best model would be $m\epsilon$ with simple composition. The  method introduced by the paper, however, builds on prior work by Liu&Talwar and improves this to become $2\epsilon$, through randomly choosing and running hyper parameter settings.

**Summary Of The Review:**

I don't see any major issues with the paper and I find the problem addressed very relevant.

---

> ### Author Response · Authors · 2021-11-17
> **Response to reviewer CBQT**
>
> We would like to thank the reviewer for their time and feedback. In the following, we respond inline to the reviewer’s questions.
>
> >_**1. How would this method be extended to be used with adaptive search methods? Grid search can be too time consuming/inefficient, especially when used with DP-SGD (as DP-SGD needs per-example gradients that are expensive to obtain) and there are some reinforcement learning based methods that can yield optimal hyper parameters much faster than grid search. Can this method be modified, maybe with a higher privacy expenditure, to be used in those cases?**_
>
> This is a good question. Our analysis does not support adaptively choosing the hyperparameter candidates based on the outcomes of prior trials. It seems likely that an arbitrary adaptive hyperparameter search would not admit any better analysis than composition. We could consider whether some particular RL-based approach could be analyzed and would appreciate it if the reviewer has any suggestions. That said, we do see certain refinements to naive hyperparameter tuning that are worth considering. One example is early stopping, which is commonly used in hyperparameter tuning to avoid wasting computational resources when the first training epochs of specific trials indicate they will not provide a competitive outcome. We thank the reviewer for highlighting this; we added a discussion of it to the conclusion of our revised manuscript.
>
> >_**2. What kind of real-life attack scenario would you envision that could actually use the best models obtained by hyper parameter tuning to extract information. Basically, what type/how much information about a model do the set of optimal hyper parameters leak?**_
>
> Intuitively, hyperparameter choices leak increasingly more information as the search space grows: in theory at least, an exponentially-large hypothesis space could leak arbitrary information about the training set. For a realistic search space, information about only a few unlucky individuals would be leaked, but this could happen in surprising ways which are hard to predict. Our intuition is that it is outliers who are most vulnerable, as they are most likely to singularly affect hyperparameter choices. However, in general differential privacy degrades gracefully, so the hyperparameter search does need to be abused for it to leak significant amounts of private information and our positive results show that, with a little bit of care, we can provably thwart privacy attacks even for large hyperparameter searches.

---

> > ### Comment · Reviewer_CBQT · 2021-11-22
> > **Response**
> >
> > Thanks for the author response, my questions have been addressed.

---

> > ### Author Response · Authors · 2021-11-22
> > **Adaptive Search Methods**
> >
> > The reviewer asked about adaptive hyperparameter search methods. We recently became aware of an independent paper which studies this question entitled [The Role of Adaptive Optimizers for Honest Private Hyperparameter Selection](https://openreview.net/pdf/66b8c252a9cd76c64ac9e620dd9768a180ed47c9.pdf).
> > This new paper analyses adaptive search methods using composition results and compares them with the results of Liu & Talwar (2019), which are for non-adaptive search (a.k.a. "grid search") like our results. We have added a citation in our submission.

---

### Decision · Program_Chairs · 2022-01-20

**Decision:**

Accept (Oral)

**Comment:**

This paper tackles a problem at the intersection of AutoML and trustworthiness that has not been studied much before, and provides a first solution, leaving much space for a lot of interesting future research.
All reviewers agree that this is a strong paper and clearly recommend acceptance.
I recommend acceptance as an oral since the paper opens the door for a lot of interesting follow-ups.